# InterControl: Zero-shot Human Interaction Generation by Controlling Every Joint

**Zhenzhi Wang**[1], **Jingbo Wang**[2], **Yixuan Li**[1], **Dahua Lin**[1,2], **Bo Dai**[3,2]
[1]The Chinese University of Hong Kong, [2]Shanghai Artificial Intelligence Laboratory,
[3]The University of Hong Kong
{wz122,ly122,dhlin}@ie.cuhk.edu.hk, wangjingbo@pjlab.org.cn
bdai@hku.hk

## Abstract

Text-conditioned motion synthesis has made remarkable progress with the emergence of diffusion models. However, the majority of these motion diffusion models are primarily designed for a single character and overlook multi-human interactions. In our approach, we strive to explore this problem by synthesizing human motion with interactions for a group of characters of any size in a zero-shot manner. The key aspect of our approach is the adaptation of human-wise interactions as pairs of human joints that can be either in contact or separated by a desired distance. In contrast to existing methods that necessitate training motion generation models on multi-human motion datasets with a fixed number of characters, our approach inherently possesses the flexibility to model human interactions involving an arbitrary number of individuals, thereby transcending the limitations imposed by the training data. We introduce a novel controllable motion generation method, InterControl, to encourage the synthesized motions maintaining the desired distance between joint pairs. It consists of a motion controller and an inverse kinematics guidance module that realistically and accurately aligns the joints of synthesized characters to the desired location. Furthermore, we demonstrate that the distance between joint pairs for human-wise interactions can be generated using an off-the-shelf Large Language Model (LLM). Experimental results highlight the capability of our framework to generate interactions with multiple human characters and its potential to work with off-the-shelf physics-based character simulators. Code is available at https://github.com/zhenzhiwang/intercontrol.

## 1 Introduction

Generating realistic and diverse human motions is a vital task in computer vision, as it has diverse applications in VR/AR, games, and films. In recent years, great progress has been achieved in human motion generation by introducing VAE [31], Diffusion Model [23, 53] and large language models [5]. These methods commonly investigated single-person motion generation given texts or action classes [15, 14, 46, 71, 55, 6, 13, 45], part of motion [10, 19, 55], or other related modalities [35, 34, 56, 3, 18], yet overlooked multi-person interactions. By naively putting their generated single-person motions in a shared global space, such motions could easily penetrate each other. They cannot even perform simple interactions like handshaking due to lack of the ability to control two people's hands to reach the same location at the same time. Many multi-person datasets [1, 16, 42, 59] lacks text annotations and focus on motion completion given prefix motions. Recently, InterGen [36] collected a two-person interaction generation dataset, and let model to learn two-person motions from data. It is limited by the fixed number of characters and cannot generalize to arbitrary numbers. Previous methods commonly ignore a good design for general interaction modeling.

38th Conference on Neural Information Processing Systems (NeurIPS 2024).

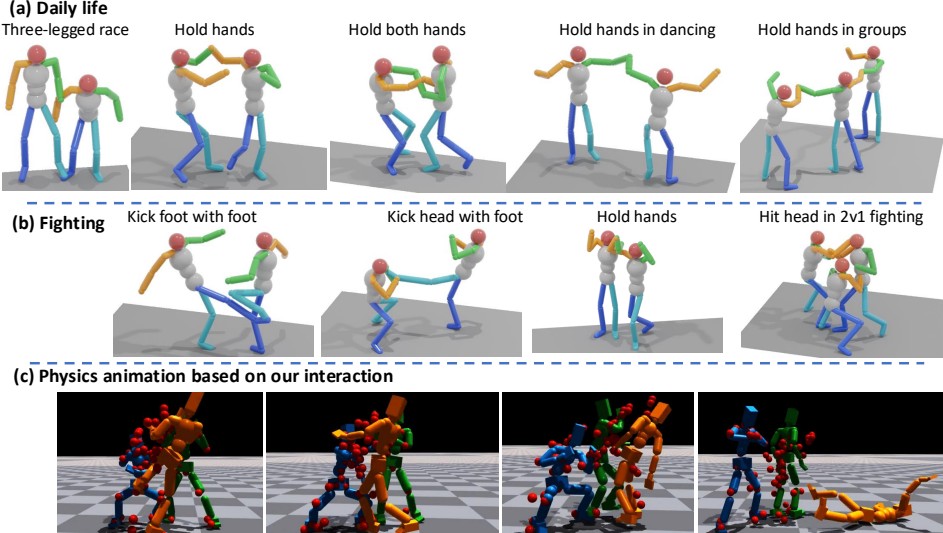

Figure 1: InterControl is able to generate **interactions of a group of people** given joint-joint **contact** or **separation** pairs as spatial condition, and it is only trained on **single-person data**. Our generated interactions are realistic and similar to real interactions in internet images in (a) daily life and (b) fighting. (c) shows our generated group motions (red dots) could serve as reference motions for physics animation.

This paper investigates a special yet widely used form of human interactions: interactions that could be quantitatively described by spatial relations of human joints, such as distances or orientations, as shown in Fig. 1 (a) and (b). Such interactions are conceptually simple, as their semantics are almost from spatial relations. Thus, they do not require additional interaction data. It only needs pretrained models from single-person data and could be generalized to an arbitrary number of humans. We define *human interactions* as steps of *joint-joint contact pairs* and devise a single-person motion generation model to take such contact pairs as control signals. Besides, orientations could also be used in control, such as making two people face each other. In this way, interaction generation is transformed to controllable motion generation. Inspired by [64], we adapt descriptions of interactions as joint contact pairs by leveraging Large Language Models (LLMs). Thus, human interactions are annotation-free, and interactions could also involve multiple human joints.

As interactions are adapted to our defined joint contact pairs, the key challenge to generate interactions is the *precise spatial control* to satisfy the constraint of spatial controls. This difficulty lies in two parts: (1) the discrepancy between *control signals in global space* and *relative motion representation* in mainstream pretrained models [14, 55]: As semantics of motions are independent to global locations, previous works [14, 55] commonly utilize the relative motions, where global locations could only be inferred by aggregating velocities. It poses challenges to control local human poses with global conditions. Previous attempts [55, 51] exploit the inpainting ability of a pretrained model, yet they are unable to control global joints. GMD [27] proposes a two-stage model of separated root trajectory generation and local pose generation. Although it manages to control root positions, controlling *every joint at any time* is still infeasible. (2) the *sparse* control signals in the motion sequence: Control signals could be sparse in both temporal and joint dimension, model needs to adaptively adjust trajectories in uncontrolled frames to satisfy the intermittent constraints.

In this paper, we propose InterControl, a novel human interaction generation method that is able to precisely control the position of any joint at any time for any person, and it is only trained on single-person motion data. By adding spatial controls to MDM [55], InterControl is a unified framework of two types of spatial control modules: (1) *Motion ControlNet* inspired by ControlNet [70]: It is initialized from a pretrained MDM [55] and takes global spatial locations as input for joint control in the global space. It is able to generate coherent and high-fidelity motions yet joint positions in global space are not perfect. (2) *Inverse Kinematics (IK) Guidance* for joint locations: To further align generated motions and spatial conditions precisely, we use inverse kinematics (IK) [44] to guide the denoising steps towards desired positions. It could be regarded as a classifier guidance [9],

yet it has no extra classifiers. We utilize L-BFGS [37] as the optimizer to directly align the global conditions in the local space. With two proposed modules, InterControl is able to control multiple joints of any person at any time. Furthermore, InterControl is able to jointly optimize multiple types of spatial controls, such as orientation alignment, collision avoidance, and joint contacts, as long as the distance measures in IK guidance are differentiable. By exploiting its joint control ability, our model is able to generate multi-person interactions with rich contacts, where no multi-person interaction datasets are needed. Our generated interactions could further serve as the reference motion to generate physical animation with meaningful human-wise reactions in simulators. As shown in Fig. 1 (c), one character could actually hit down the other with his fists by taking our generated fighting motions as input. Extensive experiments in HumanML3D [14] and KIT-ML [47] datasets quantitatively validates our joint control ability, and the user study on generated interactions shows a clear preference over previous methods.

To summarize, our contributions are twofold: (1) We are the first to generate multi-person interactions with a single-person motion generation model in a zero-shot manner. (2) We are the first to perform precise spatial control of every joint in every person at any time for interaction generation.

## 2    Related Work

### 2.1    Human Motion Generation

Synthesizing human motions is a long-standing topic. Previous efforts integrate extensive multimodal data as condition to facilitate conditional human motion generation, including text [15, 14, 46, 71, 55, 6, 30], action label [13, 45], part of motion [10, 19, 55], music [35, 34, 56], speech [3, 18] and trajectory [49, 27, 28]. As texts are free-form information that convey rich semantics, recent progress in motion generation are mainly based on text conditions. For example, FLAME [30] introduces transformer [58] to process variable-length motion data and language description. MDM [55] introduces the diffusion model and uses classifier-free guidance for text-conditioned motion generation. MLD [6] further incorporates a VAE [31] to encode motions into vectors and makes the diffusion process in the latent space. Physdiff [68] integrates physical simulators as constraints in the diffusion process to make the generated motion physically plausible and reduce artifacts. PriorMDM [51] treats pretrained MDM [55] as a generative prior and controls MDM by motion inpainting. Our InterControl also use a pretrained MDM, yet we further train a Motion ControlNet instead of using inpainting. A concurrent work OmniControl [65] also incorporate classifier guidance [9] and controlnet [70] modules to control all joints in MDM, yet it focuses on single-person motion generation and does not investigate human interaction generation.

### 2.2    Human-related Interaction Generation.

As human motions could be affected or interacted by surrounding humans [72, 29, 57], objects [66, 54, 12, 33, 26] and scenes [62–64, 73, 20, 61], generating interactions is also an important topic. Previous methods are mainly about human-scene/object interaction. For example, Interdiff [66] uses the contact point of human joints and objects as the root to generate object motions. UniHSI [64] exploits LLM to generate contact steps between human joints and scene parts as an action plan and control the agent perform the plan via reinforcement learning. As previous human-human interactions datasets [42, 59] only contains very few multi-person sequences, previous human-human interaction methods [60, 67] are mainly limited to unsupervised motion completion without texts. Recently, InterHuman dataset [36] is proposed for text-conditioned multi-person interaction generation, yet it only consider the two-person situation and is not able to model more people's interaction. To the best of our knowledge, we are the first to enable a single-person text-conditioned motion generation model to perform interactions between a group of people by controlling diverse joints of each person.

### 2.3    Controllable Diffusion Models

Diffusion-based generative models have achieved great progress in generating various modalities, such as image [50, 22, 9, 53], video [11, 17, 24] and audio [32]. Conditions and controlling ability in diffusion models are also well studied: (1) Inpainting-based methods [8, 7] predict part of the data with the observed parts as condition and rely on diffusion model to generate consistent output, which is used in PriorMDM [51]. (2) Classifier-guidance [9] trains a separate classifier and exploits the

gradient of classifier to guide the diffusion process. Our InterControl inherits the spirit of classifier-guidance, yet our guidance is provided by Inverse Kinematics (IK) and no classifier is needed. (3) Classifier-free guidance [22] trains a conditional and an unconditional diffusion model simultaneously and trade-off its quality and diversity by setting weights. (4) ControlNet [70] introduces a trainable copy of pretrained diffusion model to process the condition and freezes the original model to avoid degeneration of generation ability. It enables diverse types of dense control signals for various purpose with minimal finetuning effort. Our InterControl also incorporate the idea of ControlNet [70] to finetune the pretrained MDM [55] to process spatial control signals and improve the quality of generated motions after joint control.

## 3   InterControl

InterControl aims to generate interactions with only single-person motion data by precisely controlling every joint of every person at any time, conditioned on text prompts and joint relations. We first formulate interaction generation in Sec. 3.1, and then introduce control modules for a single-person motion diffusion model in Sec. 3.3 and Sec. 3.4. Finally we show details to generate interactions from our model in Sec. 3.5.

### 3.1   Formulation of Interaction Generation

Inspired by human-scene interaction [64], we define human interactions as joint contact pairs $\mathcal{C} = \{\mathcal{S}_1, \mathcal{S}_2, \ldots\}$, where $\mathcal{S}_i$ is the $i^{th}$ contact step. Taking two-person interaction as an example, each step $\mathcal{S}$ has several contact pairs $\mathcal{S} = \left\{\left\{j_1^1, j_1^2, t_1^s, t_1^e, c_1, d_1\right\}, \left\{j_2^1, j_2^2, t_2^s, t_2^e, c_2, d_2\right\}, \ldots\right\}$, where $j_k^1$ is the joint of person 1, $j_k^2$ is the joint of person 2, $t_k^s$ and $t_k^e$ means the start and end frame of the interaction, $c_k$ means contact type from {contact, avoid} to pull or push the joint pairs, $d_k$ is the desired distance in the interaction. By converting the contact pairs $\mathcal{S}$ to the mask $m$ and distance $d$, and taking others' joint positions as condition, we could guide the multi-person motion generation process to interact between joints in the form of spatial distance. In this way, interaction generation is transformed to be controllable single-person motion generation taking a text prompt $p$ and a spatial control signal $c \in \mathbb{R}^{N \times J \times 3}$ as input. Its goal is to predict motion sequence $x \in \mathbb{R}^{N \times D}$ whose joints in the global space is aligned with spatial control $c$, where $N$ is number of frames, $J$ is number of joints (e.g., 24 in SMPL [38]), and $D$ is the dimension of relative joint representations (e.g., 263 in HumanML3D [14]). Incorporating spatial control in motion generation presents challenges due to the discrepancy between relative motion representation $x$ and global $c$.

### 3.2   Human Motion Diffusion Model (MDM)

**Relative Motion Representation.** HumanML3D [14] dataset proposes a widely-used [55, 68, 51, 6] relative motion representation, and is proved to be easier to learn realistic motions, as the semantics of human motion is independent of global positions. It consists of root joint velocity, other joints' positions, velocities and rotations in the root space, and foot contact labels. To convert it to the global space, root velocities are aggregated, then other joints will be computed based on root. Please refer to Sec. 5 of HumanML3D [14] for details. Due to such discrepancy, previous inpainting-based methods [55, 51] is not able to control MDM in global space. GMD [27] decouples motion generation to two separated generation process of root trajectory and pose relative to root, yet it can only control root joint. Directly adopting global joint positions to generate motions yields unnatural human poses, such as unrealistic limb lengths.

**Diffusion Process in MDM.** Motivated by the success of image diffusion models [22, 50, 70, 9, 53], Motion Diffusion Model (MDM) [55] is proposed to synthesize sequence-level human motions conditioned on texts $p$ via classifier-free guidance [22]. The diffusion process is modeled as a noising Markov process $q\left(x_t \mid x_{t-1}\right) = \mathcal{N}\left(\sqrt{\alpha_t}x_{t-1}, (1 - \alpha_t)\,I\right)$, where $\alpha_t \in (0, 1)$ are small constant hyper-parameters, thus $x_T \sim \mathcal{N}(0, I)$ if $\alpha_t$ is small enough. Here $x_t \in \mathbb{R}^{N \times D}$ is the entire motion sequence at denoising time-step $t$, and there are $T$ time-steps in total. Thus, $x_0$ is the clean motion sequence, and $x_T$ is a random noise to be sampled. The denoising Markov process is defined as $p_\theta\left(x_{t-1} \mid x_t, p\right) = \mathcal{N}\left(\mu_\theta(x_t, t, p), (1 - \alpha_t)\,I\right)$, where $\mu_\theta(x_t, t, p)$ is the estimated posterior mean for the $t - 1$ step from a neural network based on the input $x_t$ and $\theta$ is its parameters. Following MDM, we predict the clean motion $x_0(x_t, t, p; \theta)$ instead of the noise $\epsilon$ via a transformer [58], and the posterior mean $\mu_\theta(x_t, t, p)$ is

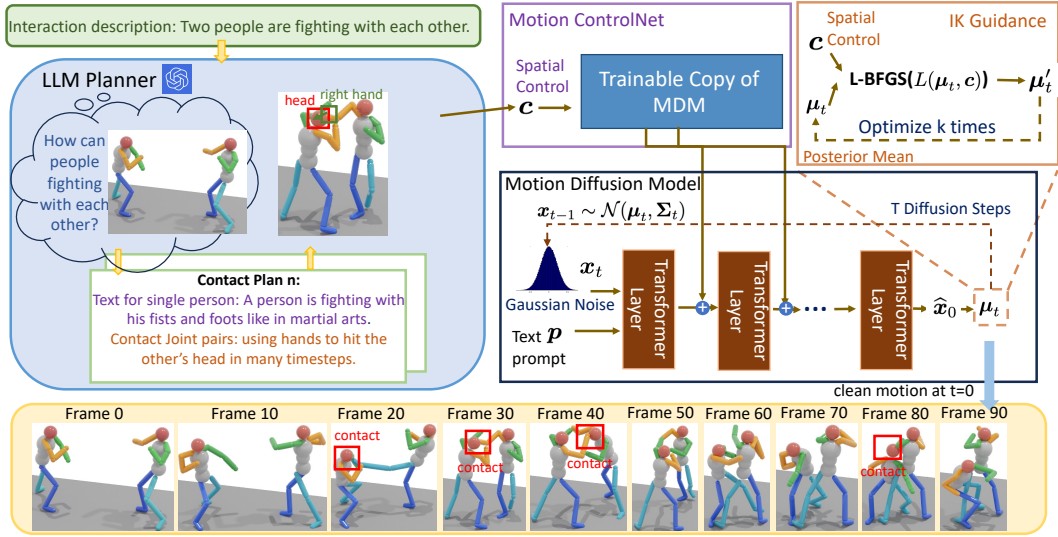

Figure 2: **Overview.** Our model could precisely control human joints in the global space via the Motion ControlNet and IK guidance module. By leveraging LLM to adapt interaction descriptions to joint contact pairs, it could generate multi-person interactions via a single-person motion generation model in a zero-shot manner.

$$\boldsymbol{\mu}_\theta(\boldsymbol{x}_t, t, \boldsymbol{p}) = \frac{\sqrt{\bar{\alpha}_{t-1}}\beta_t}{1-\bar{\alpha}_t}\boldsymbol{x}_0(\boldsymbol{x}_t, t, \boldsymbol{p}; \theta) + \frac{\sqrt{\alpha_t}(1-\bar{\alpha}_{t-1})}{1-\bar{\alpha}_t}\boldsymbol{x}_t, \tag{1}$$

where $\beta_t = 1 - \alpha_t$ and $\bar{\alpha}_t = \prod_{s=0}^t \alpha_s$. MDM's parameter $\theta$ is trained by minimizing the $\ell_2$-loss $\|\boldsymbol{x}_0(\boldsymbol{x}_t, t, \boldsymbol{p}; \theta) - \boldsymbol{x}_0^*\|_2^2$ where $\boldsymbol{x}_0^*$ is the ground-truth motion and $\boldsymbol{x}_0(\boldsymbol{x}_t, t, \boldsymbol{p}; \theta)$ is MDM's prediction of $\boldsymbol{x}_0$ at denoising timestep $t$.

### 3.3 Motion ControlNet for MDM

As MDM is initially conditioned on texts $\boldsymbol{p}$, it requires fine-tuning to accommodate spatial conditions $\boldsymbol{c}$. This is challenging due to the potential sparsity of $\boldsymbol{c}$ across temporal and joint dimensions: (1) Control may be required for only a few joints, necessitating adaptive adjustment of the remaining joints to preserve realistic motion. (2) Control may be desired for only a select few frames, thus the model must interpolate natural human motions for the rest of the sequence.

Inspired by ControlNet [70], we introduce Motion ControlNet to generate realistic and high-fidelity motions guided by condition $\boldsymbol{c}$. It is a trainable copy of MDM, while MDM is frozen in our training process. Each transformer encoder layer in ControlNet is connected to its MDM counterpart via a zero-initialized linear layer. This allows InterControl to commence training from a state equivalent to a pretrained MDM, acquiring a residual feature for $\boldsymbol{c}$ in each layer through back-propagation. To process $\boldsymbol{c}$, the uncontrolled joints, frames, and XYZ-dim are masked as 0. We find that the vanilla $\boldsymbol{c} \in \mathbb{R}^{N \times 3J}$ is effective enough to control the pelvis (root) joint, yet it is still sub-optimal for other joints. Thus, we design a relative condition indicating the distance from the current positions of each joint to $\boldsymbol{c}$. Suppose $R(\cdot)$ is a forward kinematics (FK) to convert relative motion $\boldsymbol{x} \in \mathbb{R}^{N \times D}$ to global space $R(\boldsymbol{x}) \in \mathbb{R}^{N \times J \times 3}$, the relative condition is $\boldsymbol{c}' = \boldsymbol{c} - R(\boldsymbol{x})$. To provide additional clues, we also use $\boldsymbol{c}'' = \boldsymbol{c} - R(\boldsymbol{x})^{root}$ to represent the distance from the current root to the desired position. We also use the normal of triangles (pelvis, left/right shoulder) $\boldsymbol{n}^s$ and (pelvis, left/right hip) $\boldsymbol{n}^h$ to represent the current orientation of human. The final condition passed to ControlNet is $\boldsymbol{c}^{final} = (\boldsymbol{c}'||\boldsymbol{c}''||\boldsymbol{n}^s||\boldsymbol{n}^h)$, where $||$ is concatenation. Please refer to Appendix A.2 for more details.

**Network Training.** Motion ControlNet is the only part that needs finetuning in our framework, while IK guidance is an optimization method in the test time and the LLM in our framework is an off-the-shelf GPT-4 [43]. We adopt the standard ControlNet [70] training strategy, and the only difference is the data format: we first convert the relative motion to be global locations by FK, and then use random masks that keeps part of global joints to be non-zero as spatial control signals. The training objective is identical to MDM. The spatial conditions are randomly sampled in the temporal or joint dimension. The training data is single-person data only, e.g., HumanML3D [14].

## 3.4 Inverse Kinematics (IK) Guidance

While Motion ControlNet can adapt joint positions according to sparse conditions, the alignment between predicted poses and global spatial conditions often lacks precision. As Inverse Kinematics (IK) is a classic method for optimizing joint rotations to achieve specific global positions, we employ it to guide the diffusion process towards spatial conditions at test time in a classifier guidance [9] manner, named IK guidance.

**IK Guidance on general form of losses.** Inspired by classifier guidance [9] and loss-guided diffusion [52], we employ losses in the global space to steer the denoising process. IK guidance accommodates various forms of distance measurements, enabling both minimization and maximization for flexible control over joint interactions, such as attraction or repulsion. Given the global position $\boldsymbol{c} \in \mathbb{R}^{N \times J \times 3}$, the distance between a joint and condition is $\boldsymbol{d}_{nj} = \|\boldsymbol{c}_{nj} - R(\boldsymbol{\mu}_t)_{nj}\|_2$, where $\boldsymbol{\mu}_t$ is short for $\boldsymbol{\mu}_\theta(\boldsymbol{x}_t, t, \boldsymbol{p})$ mentioned in Sec. 3.2, and $R(\cdot)$ is forward kinematics (FK). To allow the interaction of joints with some given distances $d' \in \mathbb{R}^{N \times J \times 3}$, loss of one joint is $\boldsymbol{l}_{nj} = \text{ReLU}\left(\boldsymbol{d}_{nj} - d'_{nj}\right)$ to make the joint and condition be **contacted** within distance $d'_{nj}$; and it is $\boldsymbol{l}_{nj} = \text{ReLU}\left(d'_{nj} - \boldsymbol{d}_{nj}\right)$ to make the joint and condition be **far away**, where ReLU is a function to keep values $\geq 0$ and set values $\leq 0$ to 0. Finally, with a binary mask $\boldsymbol{m} \in \{0, 1\}^{N \times J \times 3}$, the total loss for all joints and frames is

$$L(\boldsymbol{\mu}_t, \boldsymbol{c}) = \frac{\sum_n \sum_j \boldsymbol{m}_{nj} \cdot \boldsymbol{l}_{nj}}{\sum_n \sum_j \boldsymbol{m}_{nj}}, \qquad (2)$$

As $\ell_2$-loss and FK are highly differentiable, we optimize $L(\boldsymbol{\mu}_t, \boldsymbol{c})$ in Equ. 2 w.r.t $\boldsymbol{\mu}_t$ using the second-order optimizer L-BFGS [37], which is commonly used in Inverse Kinematics, rather than first-order gradient methods. Classifier guidance [9] utilizes a pre-trained image classifier to direct the diffusion towards a target image class by the gradient $\nabla_{\boldsymbol{x}_t} \log f_\phi(y \mid \boldsymbol{x}_t)$, where $f_\phi$ is the classifier, $y$ is image class. Unlike this method, we do not rely on a large neural network classifier. L-BFGS has been demonstrated to better align global positions and offer quicker convergence than first-order methods. We update the posterior mean $\boldsymbol{\mu}_t$ using L-BFGS for $k$ iterations at each denoising step, where $k$ is a hyper-parameter. This optimization facilitates both pull and push types of IK guidance, corresponding to two contact types in our interaction model. To maintain consistency in data distribution between training and inference, we also apply IK guidance when training ControlNet. Additionally, employing IK guidance on $\boldsymbol{x}_0$ eliminates the need for training Motion ControlNet, thus enhancing training efficiency. In practice, using L-BFGS on both $\boldsymbol{x}_0$ and $\boldsymbol{\mu}_t$ can yield satisfactory joint and spatial condition alignment. Detailed algorithm for interaction generation is presented in Appendix A.1.

As the root position at frame $n$ is derived from cumulative root velocities up to frame $n$ in FK, a single condition at frame $n$ can influence all preceding root positions. This effect also extends to non-root joints, as their global positions are calculated from the root. Consequently, IK guidance can adaptively modify velocities from the start to frame $n$ to meet the condition at frame $n$. Moreover, IK guidance can control any combination of human joints, frames or XYZ-dims, such as controlling the left hand and right foot at a specific frame $n$.

## 3.5 Interaction Generation

Inverse Kinematics (IK) guidance can optimize various distance measures to facilitate interactions such as avoiding obstacles, preventing collisions, facilitating face-to-face engagements, or enabling joint contacts between individuals. This method allows for intricate interactions among any human joints for an indefinite number of people, despite being trained exclusively on single-person data. As delineated in Section 3.1, we characterize interactions as pairs of contacting joints. A notable feature of our IK guidance in generating interactions is that both terms of the IK guidance loss function are predicted, allowing for simultaneous optimization within a single process. Specifically, the single-person loss $L_{single}(\boldsymbol{\mu}_t, \boldsymbol{c})$ transforms into $L_{multi}(\boldsymbol{\mu}_t^a, \boldsymbol{\mu}_t^b)$ for interactions, where $a$ and $b$ represent two individuals. The L-BFGS optimizer concurrently optimizes both participants by minimizing $L_{multi}(\boldsymbol{\mu}_t^a, \boldsymbol{\mu}_t^b)$, with $\boldsymbol{\mu}_t^a$ and $\boldsymbol{\mu}_t^b$ being the respective joints engaged in interaction. Beyond distance measures, our IK guidance can optimize orientation measures as well. For example, one can calculate a person's orientation through the spatial relationship of their joints, like the cross-product of vectors from the left shoulder to the right and from the pelvis to the head. By setting two individuals' unit orientation vectors to 0, they can face each other or turn away. To ensure they face each other, we can further adjust the relation between one person's orientation vector and the vector from their head to

the other's. Such orientation relationships are vital for producing realistic interactions when we only exploit single-person motion generation ability and can be easily expanded to include larger groups. Another useful strategy in IK guidance is to prevent collision through joint separation pairs, ensuring that the torso joints of two people (such as pelvis, hips, and spines) maintain a certain distance, thereby reducing the likelihood of collisions when other joints are in contact. Besides, we can also regulate the motion region by confining the root joints within the XZ-plane using IK guidance. For the PyTorch-like code illustrating loss functions that enforce joint contacts, separations, or orientation alignment, please refer to Appendix A.1 for details.

In our framework, interaction generation is realized by using joint-joint contact pairs as control signals. These pairs can be manually crafted by users to create desired interactions, akin to utilizing ControlNet [70] in image generation. However, manually constructing joint contact pairs can be tedious, so we employ an automatic off-the-shelf GPT-4 [43] as a planner. GPT-4 infers text prompts that describe the actions of multiple people, $p^{multi}$, and converts them into single-person prompts, $p$, and contact plans, $\mathcal{C}$, through prompt engineering. The inputs for the LLM Planner include the multi-person sentences $p^{multi}$, background scenario details $\mathcal{B}$, human joint data $\mathcal{J}$, and predefined instructions, rules, and examples. Specifically, $\mathcal{B}$ encompasses the number of individuals, total motion sequence frames, and video playback speed; $\mathcal{J}$ contains names of all joints (for example, the 22 joint names in HumanML3D [14]); and the rules outline the joint contact pair format and guide the LLM to generate feasible contacts and timesteps. Our method leverages the pre-trained capabilities of GPT-4 to comprehend human joint relationships from interaction descriptions via prompt engineering without any fine-tuning. Thus, the inference process of our model is not related to LLMs, making our comparison with other methods be fair. Please refer to Appendix A.3 for details of prompts and contact plans.

## 4 Experiments

**Datasets.** We conduct experiments on HumanML3D [14] and KIT-ML [47] following MDM [55]. HumanML3D contains 14,646 high-quality human motion sequences from AMASS [41] and HumanAct12 [13], while KIT-ML contains 3,911 motion sequences with more noises.

**Evaluation Protocol.** We adopt metrics suggested by *Guo et. al.* [14] to evaluate the quality of alignment between text and motion, which are Frechet Inception Distance (**FID**), **R-Precision**, and **Diversity**. We also report metrics related to spatial controls following GMD [27] on HumanML3D dataset, which are **Foot skating ratio**, **Trajectory error**, **Location error** and **Average error**. Please refer to Appendix B.5 or papers [14, 27] for more details.

Due to the page limit, we put the implementation details and text-to-motion generation in the Appendix B.1 and B.2.

### 4.1 Single-Person Controllable Motion Generation

In Tab. 1, we compare InterControl with other spatially controllable methods [51, 27, 65]. We also include results of MDM [55] to show the controlling metrics [27] without spatial control. MDM's trajectory can significantly deviate from the intended path in the absence of control signals, with an average error often exceeding 1m. In contrast, inpainting-based control, unaware of global spatial information, results in considerable divergence, as seen with PriorMDM [51]. GMD [27] decouples this problem and generates root trajectories in the global space, so it achieves better performance in spatial control metrics. However, its limitation to only the root joint constrains its spatial control and interaction capabilities. Our InterControl could achieve very small errors in spatial control metrics for all-joint control thanks to the power of Inverse Kinematics and L-BFGS optimizer. Meanwhile, Motion ControlNet could ensure the motion data is still in the same distribution with the training set by adapting to the posterior mean updated by IK guidance in its training stage, leading to even better FID than previous methods. It is worth noting that we only use a single model to learn the control strategy for all joints, while previous method [51] needs to train separate models and blend them for multiple joints. Our method achieves similar performance with controlling one joint when extending it to control multiple joints (last two rows in Tab. 1). Compared to the recent concurrent work [65], we achieve significantly better FID and Traj./Loc. errors than it in both root joint control or random joint control. It [65] also shows a notable gap between two form of joint controls (0.310 vs. 0.218), while our method is more robust to joint variants (0.178 vs. 0.159) thanks to our special designs

Table 1: **Spatial control** results on HumanML3D [14]. → means closer to real data is better. *Random One/Two/Three* reports the average performance over 1/2/3 randomly selected joints in evaluation. † means our evaluation on their model.

| Method | Joint | FID ↓ | R-precision ↑ (Top-3) | Diversity → | Foot skating ratio ↓ | Traj. err. ↓ (50 cm) | Loc. err. ↓ (50 cm) | Avg. err.↓ (m) |
|---|---|---|---|---|---|---|---|---|
| Real data | - | 0.002 | 0.797 | 9.503 | 0.0000 | 0.0000 | 0.0000 | 0.0000 |
| MDM [55] | No Control | 0.544 | 0.611 | 9.446 | 0.0943 | 0.8909 | 0.6015 | 1.1843 |
| PriorMDM [51]† | | 0.498 | 0.586 | 9.167 | 0.0924 | 0.3726 | 0.2210 | 0.4552 |
| GMD [27]† | Root | 0.276 | 0.655 | 9.245 | 0.1108 | 0.0987 | 0.0356 | 0.1457 |
| OmniControl [65] | | 0.218 | 0.687 | 9.422 | **0.0547** | 0.0387 | 0.0096 | **0.0338** |
| Ours | | **0.159** | 0.671 | 9.482 | 0.0729 | **0.0132** | **0.0004** | 0.0496 |
| OmniControl [65] | Random one | 0.310 | **0.693** | **9.502** | 0.0608 | 0.0617 | 0.0107 | 0.0404 |
| Ours | | 0.178 | 0.669 | 9.498 | 0.0968 | 0.0403 | 0.0031 | 0.0741 |
| Ours | Random two | 0.184 | 0.670 | 9.410 | 0.0948 | 0.0475 | 0.0030 | 0.0911 |
| Ours | Random three | 0.199 | 0.673 | 9.352 | 0.0930 | 0.0487 | 0.0026 | 0.0969 |

Table 2: Evaluation on (left) spatial errors and (right) user preference in interactions.

| Spatial Errors | Traj. err. (20 cm) ↓ | Loc. err. (20 cm) ↓ | Avg. err. (m) ↓ |
|---|---|---|---|
| PriorMDM [51] | 0.6931 | 0.3487 | 0.6723 |
| Ours | **0.0082** | **0.0005** | **0.0084** |

| User-study | Preference |
|---|---|
| PriorMDM [51] | 18.8% |
| Ours | **81.2%** |

of more inputs in Motion ControlNet. Its R-precision and foot-skating ratio are slightly better than ours, we believe the reason is that their 1-st order optimization tolerates more errors when the joint alignment is hard. It is also supported by their worse Traj./Loc. yet better Avg. err., i.e., their method shows more outliers with large errors. However, their design need much more times of optimization compared to ours (e.g., 100 vs. 5) and leads to longer inference time than ours (120s vs. 80s).

## 4.2 Zero-Shot Multi-Person Interaction Generation

To validate our model's interaction generation ability, we analyze the spatial control results in interaction scenarios and perform an user study to qualitatively compare our model with PriorMDM [51]. We also introduce an potential application of our interaction generation method for physics animation.

**Spatial Control.** In Tab. 2 (left), we compare spatial-related metrics with PriorMDM in zero-shot human interaction generation. Specifically, we collect 100 descriptions of two-person actions from InterHuman Dataset [36] and let an off-the-shelf GPT-4 [43] to adapt them to single-person motion descriptions and joint-joint contact pairs via prompt engineering (see Tab. 7 in Appendix). Then, we utilize an InterControl model pretrained on the HumanML3D dataset to generate human interactions conditioned on text prompts and joint contact pairs. The spatial-related metrics are reported over controlled joints and frames. InterControl achieves good performance of spatial errors in interaction scenarios, indicating its robustness in precise spatial control for multiple humans. In contrast, PriorMDM [51] could only take interaction descriptions as input and unable to perform spatial control, leading to much larger spatial errors.

**User Study.** We conduct a user study to qualitatively compare our method with PriorMDM on the text-conditioned two-person interaction generation. 134 unique users were participating in the user study, where each user will answer 19 single choice questions to compare our results with PriorMDM [51]. Results in Tab. 2 (right) shows that our generated interactions are clearly preferred over PriorMDM by a percent of 81.2%. We also shows an example sequence of qualitative comparison with PriorMDM [51] in the user study in Fig. 3. PriorMDM [51] shows severe torso collision between two human skeletons and the generated two-people motion is not aligned with the interaction description, while our model has no torso collision thanks to the collision avoidance loss in our IK guidance. Besides, our method also produces reasonable kicking actions between two people according to the semantics of interaction description. Please refer to Appendix B.4 for details.

**Qualitative results:** Although our model is only trained on single-person data, it is still possible to generate interactions between an arbitrary number of people via our designed format of interaction. In Fig. 4, we show two representative results of zero-shot interaction generation. (1) Two-person dancing: In addition to the single person dancing from the pretrained ability of single-person model, we further let them hold hands from time to time and prevent them from collision between their torsos. To further make their dance natural, we also employ a loss to promote their orientations to

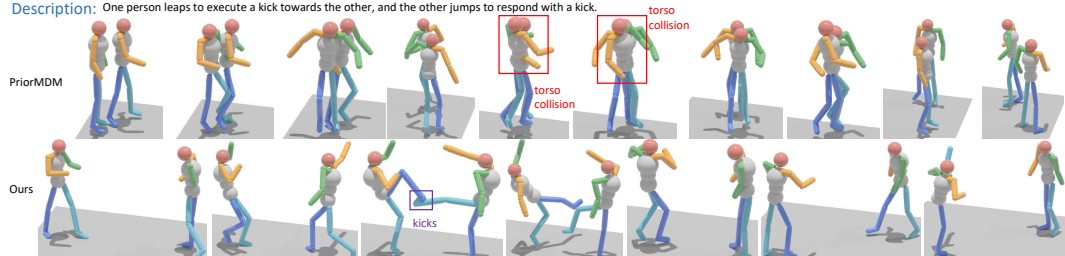

Figure 3: Comparison with PriorMDM [51] in **user-study** of zero-shot human interaction generation.

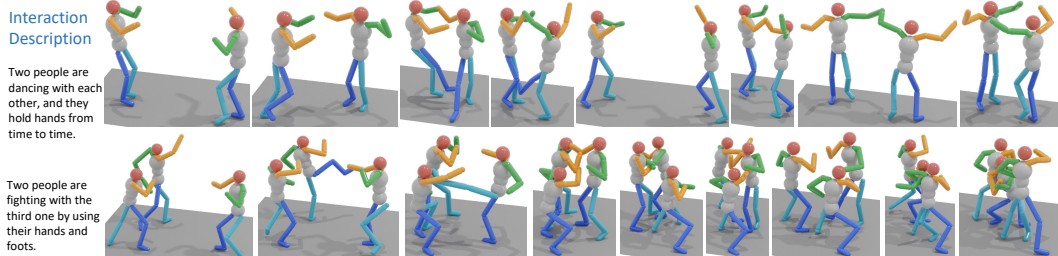

Figure 4: **Qualitative results** of zero-shot human interaction generation.

be face-to-face. (2) Three-person fighting: In addition to a single person performing punching and kicking, we further let them punch or kick others' head and torso, and also prevent their torsos from collision. Compared to existing interaction generation method [36], our method is able to generate interaction between any number of people, while InterGen [36] is only able to generate two-person interaction. Besides, our method is the first method to leverage single-person motion generation model to generate human interactions in a zero-shot manner.

**Application:** Our method is able to seamlessly integrate with off-the-shelf character simulation approaches, allowing us to synthesize physically plausible human reactions. As shown in Fig. 1 (c), our method synthesizes the motions, where the orange character is fighting with other two characters, as the reference of the SoTA physics-aware motion imitator [40]. The interactions of our motions are designed to hit heads of other characters with fists. Leveraging the precise spatial control provided by our approach, the animated characters in the simulator can accurately respond to these impacts, resulting in realistic reactions such as being knocked down. This capability to generate spatially coherent multi-human interactions enables our method to improve the plausibility and responsiveness of synthesized reactions within physics-based character animations.

## 4.3 Ablation Studies

To further investigate the effectiveness of InterControl, we ablate our method in Tab. 3 and reveal some key information in controlling the motion generation model in the global space. Then we also analyze the computational costs of our method to ensure our control is efficient. We will refer to the variants of InterControl by row numbers in Tab. 3. All experiments are trained on all joints and evaluated with randomly selected joints to report average performance.

**Motion ControlNet.** By dropping ControlNet, we find that IK guidance could still follow spatial controls with very low errors, yet the motion quality (e.g., FID) is significantly damaged (row 1 vs. row 2). Our ControlNet could adapt to the posterior distribution updated by IK guidance, and produce high-quality motion data. We also find that our $c^{final}$ provides key information in controlling all joints: For root control only, the FID of $c^{final}$ and $c$ shows small difference. However, the FID of root control is always slightly better than all-joint control ($\sim 0.07$) when we use $c$, indicating insufficient information in all-joint control. We alleviate this by introducing extra information in $c^{final}$ for Motion ControlNet and improve the FID of all-joint control from 0.227 (row 3) to 0.178 (row 1).

**IK guidance.** By dropping IK guidance, Motion ControlNet can produce good semantic-level metrics (e.g., FID) compared with MDM by using extra spatial cues (row 4). However, this variant will lead to more spatial errors and cannot strictly follow spatial controls in global space. As precise joint alignment is vital for interactions, IK guidance is important for our InterControl. Another variant is

Table 3: **Ablation studies** on the HumanML3D [14] dataset.

| Item | Method | FID ↓ | R-precision ↑ (Top-3) | Diversity → | Foot skating ratio ↓ | Traj. err. ↓ (50 cm) | Loc. err. ↓ (50 cm) | Avg. err.↓ (m) |
|---|---|---|---|---|---|---|---|---|
| (1) | Ours (random joint) | **0.178** | 0.669 | **9.498** | 0.0968 | 0.0403 | 0.0031 | 0.0741 |
| (2) | w/o ControlNet | 0.965 | 0.621 | 9.216 | 0.1624 | 0.0879 | 0.0059 | 0.1013 |
| (3) | w/ original $c$ | 0.227 | 0.656 | 9.544 | 0.1004 | 0.0697 | 0.0042 | 0.0785 |
| (4) | w/o IK guidance | 0.187 | 0.664 | 9.598 | **0.0704** | 0.8569 | 0.4553 | 0.6557 |
| (5) | IK guidance on $x_0$ | 0.211 | 0.668 | 9.394 | 0.1164 | 0.0907 | 0.0088 | 0.0981 |
| (6) | w/ 1-st order grad | 0.198 | 0.668 | 9.472 | 0.0987 | 0.0879 | 0.0096 | 0.0877 |
| (7) | sparsity = 0.25 | 0.248 | **0.671** | 9.442 | 0.0801 | 0.0106 | 0.0007 | 0.0546 |
| (8) | sparsity = 0.025 | 0.255 | 0.663 | 9.520 | 0.0705 | **0.0015** | **0.0001** | **0.0067** |

Table 4: **Inference time analysis** on a NVIDIA A100 GPU.

| Sub-Modules | MDM | + Control Module | + Guidance $t \in [10, 999]$ | + Guidance $t \in [0, 9]$ |
|---|---|---|---|---|
| Time (s) | 39.1 | 57.3 | 76.5 | 80.1 |

updating IK guidance on ControlNet's prediction $x_0$ (row 5), instead of the posterior mean $\mu_t$. Its advantage is faster training speed because IK guidance is no longer needed in training ControlNet (similar to classifier guidance [9]) yet it leads to slightly worse FID than using $\mu_t$. We believe the reason is that IK guidance still changes the data distribution in denoising steps even if it is updated on $x_0$. Finally, we also report the result of 1-st order gradient in classifier guidance [9] (row 6) instead of L-BFGS. We find it takes more computations to achieve similar performance with L-BFGS, which is analyzed below.

**Inference time analysis.** In practice, we find that IK guidance in last few denoising steps (e.g., $t \in [0, 9]$) is vital for precise joint control, while most denoising steps $t \in [10, 999]$ are less important yet take most of computations. IK guidance on $x_0$ with only once L-BFGS in $t \in [10, 999]$ and 10 times in $t \in [0, 9]$ could leads FID 0.234 in controlling all joints, yet leads to minimal extra computations. We report its total inference time of 1000 denoising steps by adding sub-modules step-by-step in Tab. 4. GMD [27] needs 110s to run two-stage diffusion models, while we only needs 80s. Gradient-based optimization in the recent work [65] needs 120s to achieve similar control quality. Leveraging GPU parallel computing capabilities, InterControl can efficiently generate motions for a batch of 32 people in 91 seconds, enabling efficient group motion generation.

**Sparse control signals in temporal.** As a key challenge of spatial control is the sparsity, we also report results with sparsely selected frames as control (sparsity = 0.25 and 0.025) in Tab. 3 (row 7 and 8). Our model demonstrates consistent performance in both spatial error and semantic-level metrics when using sparse signals, e.g., FID 0.255 and avg. err. 0.0467 with sparsity 0.025, while GMD [27] achieves FID 0.523 and avg. err. 0.139 with the same sparsity.

## 5  Conclusion and Limitations

We presented InterControl, a multi-person interaction generation method that is only trained on single-person motion data. It could generate interactive human motions of an arbitrary number of people. We achieve this by enabling a text-conditioned motion generation model with the ability to control every joint of every person at any time. We propose two complementary modules, named Motion ControlNet and IK guidance, to improve both the spatial alignment between joints and desired positions, and the overall quality of whole motions. Extensive experiments are conducted on HumanML3D and KIT-ML benchmarks to validate the effectiveness and efficiency of our proposed modules. We enable InterControl the ability of text-conditioned interaction generation by leveraging the knowledge of LLMs. Qualitative results and user study validate that InterControl could generate high-quality interactions by precise spatial joint control.

**Limitations.** As InterControl is not trained on multi-person data, its definition of interaction is based on distances (being *contacted* or *separated*) or orientations. Its motion quality is from motion generation model trained on single-person motion data, and the plausibility of interactions is from the knowledge of LLMs, i.e., to what extent the joint contact pairs are consistent to the semantics of interaction descriptions. Yet, InterControl could generate interactions of an arbitrary number of people, while all existing interaction generation methods cannot.

**Acknowledgment.** This project is funded in part by Shanghai Artificial Intelligence Laboratory, CUHK Interdisciplinary AI Research Institute, and the Centre for Perceptual and Interactive Intelligence (CPII) Ltd under the Innovation and Technology Commission (ITC)'s InnoHK. We would like to thank Tianfan Xue for his insightful discussion.

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

# Appendix

## A    More Details about InterControl

---
**Algorithm 1** Two-people interaction model inference

---
**Require:** a Motion Diffusion Model $M$ with parameter $\theta$, a Motion ControlNet $C$ with parameter $\phi$, interaction prompts $\boldsymbol{p}^{multi}$, number of L-BFGS $K$, Forward Kinematics operation FK, masked selection operation $S$.
1:   $\boldsymbol{x}_T^a, \boldsymbol{x}_T^b \sim \mathcal{N}(0, \boldsymbol{I})$
2:   **for** $t$ from $T$ to 1 **do**
3:       # LLM-Planner
4:       $\boldsymbol{p}^a, \boldsymbol{p}^b, \text{mask} \leftarrow \text{LLM}(\boldsymbol{p}^{multi})$
5:       # Copy Spatial Condition from Each Other
6:       $\boldsymbol{c}^a \leftarrow S(\text{FK}(\boldsymbol{x}_t^b), \text{mask})$
7:       $\boldsymbol{c}^b \leftarrow S(\text{FK}(\boldsymbol{x}_t^a), \text{mask})$
8:       # Motion ControlNet
9:       $\{\boldsymbol{f}\}^a \leftarrow C\left(\boldsymbol{x}_t^a, t, \boldsymbol{p}^a, \boldsymbol{c}^a; \phi\right)$
10:      $\{\boldsymbol{f}\}^b \leftarrow C\left(\boldsymbol{x}_t^b, t, \boldsymbol{p}^b, \boldsymbol{c}^b; \phi\right)$
11:      # Motion Diffusion Model
12:      $\boldsymbol{x}_0^a \leftarrow M\left(\boldsymbol{x}_t^a, t, \boldsymbol{p}^a, \{\boldsymbol{f}\}^a; \theta\right)$
13:      $\boldsymbol{x}_0^b \leftarrow M\left(\boldsymbol{x}_t^b, t, \boldsymbol{p}^b, \{\boldsymbol{f}\}^b; \theta\right)$
14:      $\boldsymbol{\mu}_t^a, \Sigma_t \leftarrow \mu\left(\boldsymbol{x}_0^a, \boldsymbol{x}_t^a\right), \Sigma_t$     # Posterior
15:      $\boldsymbol{\mu}_t^b, \Sigma_t \leftarrow \mu\left(\boldsymbol{x}_0^b, \boldsymbol{x}_t^b\right), \Sigma_t$     # Posterior
16:      **for** $k$ from 1 to $K$ **do**
17:         # IK guidance
18:         $\boldsymbol{\mu}_t^a, \boldsymbol{\mu}_t^b \leftarrow \text{L-BFGS}(L(\boldsymbol{\mu}_t^a, \boldsymbol{\mu}_t^b))$
19:      **end for**
20:      $\boldsymbol{x}_{t-1}^a \sim \mathcal{N}(\boldsymbol{\mu}_t^a, \Sigma_t)$
21:      $\boldsymbol{x}_{t-1}^b \sim \mathcal{N}(\boldsymbol{\mu}_t^b, \Sigma_t)$
22: **end for**
23: **return** $\boldsymbol{x}_0^a, \boldsymbol{x}_0^b$

---

### A.1    Pseudo-code of IK guidance

Here we elaborate the details of IK guidance's algorithm. As we mentioned in the main paper, IK guidance could be performed on predicted clean motion (i.e., $\boldsymbol{x}_0$) or posterior mean in denoising step $t$ (i.e., $\boldsymbol{\mu}_t$). In practice, we find that $\boldsymbol{x}_0$ works well in root control, and it does not require IK guidance in training Motion ControlNet, leading to faster training speed. Besides, it also requires less times of L-BFGS [37], which means faster inference speed. $\boldsymbol{\mu}_t$ leads to better FID in controlling all joints, yet it requires more times of L-BFGS [37] and also need IK guidance in training Motion ControlNet. We show the pseudo-code of InterControl in interaction generation in Algorithm 1.

### A.2    Details of Motion ControlNet

In this subsection, we elaborate the details of Motion ControlNet's architecture. Motion ControlNet is designed to adaptively generate realistic and high-fidelity motion sequences based on condition $\boldsymbol{c}$. It is a trainable copy of MDM, and each transformer encoder layer of ControlNet and the original MDM is connected by a zero-initialized linear layer, as shown in Fig. 5. The parameters in the original MDM is pretrained and frozen in the entire training process. Thus, our framework in the finetuning process starts from the weights that is equivalent to a pretrained MDM due to the zero-initialized linear layers. ControlNet will learn a residual feature for spatial control signals $\boldsymbol{c}$ in each transformer layer by the back-propagated gradients. Thus, our model is able to implicitly adjust model weights for all joints and frames based on a sparse spatial condition $\boldsymbol{c}$ by learning the spatial-level conditional distribution in addition to the semantic-level distribution.

To process condition $\boldsymbol{c}$, the uncontrolled joints, frames and XYZ-dim are masked as 0. Then we use a linear layer to project the condition $\boldsymbol{c} \in \mathbb{R}^{N \times 3J}$ to the hidden dimension of transformer layers as $\boldsymbol{c}^H \in \mathbb{R}^{N \times D^H}$, and feed $\boldsymbol{c}'$ to transformer encoder layers in ControlNet. We use a zero-initialized linear layer to link the output of each layer in ControlNet to the transformer encoder layer of pretrained and frozen MDM via a residual connection [21]. We use extra information as condition for Motion

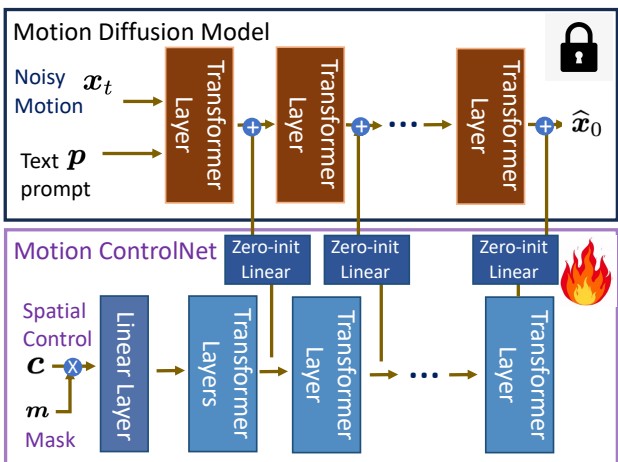

Figure 5: **Architecture of Motion ControlNet.**

ControlNet $c^{final} = cat(c', c'', n^s, n^h)$. The details of $c^{final}$ has been explained in Sec 3.3 in our main paper.

### A.3 LLM-Planner

In this section, we further elaborate the details of LLM Planner. Specifically, we collect 100 sentences describing human interactions with joint contacts from the description of InterHuman Dataset [36]. Then, we use a GPT-4 [43] with the prompt in Tab. 7 to let GPT-4 to produce joint-joint contact plans for us. For each collected sentence, we replace it as the *instruction* in the prompt, and LLM will generate 10 task plans for us, as shown in Tab. 8. We manually correct typos of task plans generated by LLM, such as typos of joint name, invalid joint name, or invalid start frame or end frame. It leads to 989 valid task plans. Finally, we write Python scripts to transform the natural language tasks plans to Python Json format, as shown in Tab. 9. We take single-person language prompts in task plans as texts for motion diffusion model, and transform information in 'steps' to joint contact masks in the spatial condition. Specifically, we update the other person's joint positions as the current person's spatial condition in each denoising step, and use the spatial condition to guide Motion ControlNet and IK guidance in the same way with single-person scenarios. We evaluate the quality of interactions by using metrics like trajectory error and average error proposed by GMD [27] in the same way with single-person scenarios. We only evaluate on joints and frames in the joint-joint contact pairs. The result on our collected 989 task plans is shown in Tab. 5 in the main paper.

## B  Additional Experiments

### B.1  Implementation Details.

We initialize parameters of both original MDM and Motion ControlNet from pretrained MDM [55] weight and freeze the parameters of original MDM during training. Following MDM [55], we use CLIP [48] model to encode text prompts. We run L-BFGS [37] in IK guidance 5 times for the first 990 denoising steps and 10 times for the last 10 denoising steps on the posterior mean $\mu_t$; and once for the first 990 steps and 10 times for the last 10 steps on clean motion $x_0$. We use IK guidance in training ControlNet when using it on $\mu_t$. We set two types of mask $m \in \{0,1\}^{N \times J \times 3}$: (1) Only keeps pelvis (root) joint for root control to fairly compare with previous methods; (2) Randomly keep one joint in each iteration to learn to control all joints for interaction generation. Each type of mask will be used in both training and inference for consistency. Thus, we get two model weights, where (1) could be fairly compared with previous methods and we use (2) for interaction generation. We use AdamW [39] optimizer and set the learning rate as 1e-5.

### B.2  Text-to-Motion Generation Results

To generally compare our InterControl with previous text-conditioned motion generation methods, we report the alignment quality of text and generated motions suggested by *Guo et. al.* [14] in Tab. 5.

Table 5: **Text-to-motion evaluation** on the (left) HumanML3D [14] and (right) KIT-ML [47] datasets. The right arrow → means closer to real data is better. Methods in the upper part are unable to perform spatial control. † means our implementation.

| HumanML3D | FID ↓ | R-precision ↑ (Top-3) | Diversity → |
|---|---|---|---|
| Real | 0.002 | 0.797 | 9.503 |
| JL2P [2] | 11.02 | 0.486 | 7.676 |
| Text2Gesture [4] | 7.664 | 0.345 | 6.409 |
| T2M [14] | 1.067 | 0.740 | 9.188 |
| MotionDiffuse [71] | 0.630 | **0.782** | 9.410 |
| MLD [6] | 0.473 | 0.772 | 9.724 |
| PhysDiff [68] | 0.433 | 0.631 | - |
| T2M-GPT [69] | **0.116** | 0.775 | 9.761 |
| MotionGPT [25] | 0.232 | 0.778 | 9.528 |
| MDM [55] | 0.544 | 0.611 | 9.446 |
| PriorMDM [51] | 0.540 | 0.640 | 9.160 |
| GMD [27] | 0.212 | 0.670 | 9.440 |
| OmniControl [65] | 0.218 | 0.687 | 9.422 |
| Our InterControl | 0.159 | 0.671 | **9.482** |

| KIT-ML | FID ↓ | R-precision ↑ (Top-3) | Diversity → |
|---|---|---|---|
| Real | 0.031 | 0.779 | 11.08 |
| T2M [14] | 3.022 | 0.681 | 10.72 |
| MotionDiffuse [71] | 1.954 | 0.739 | **11.10** |
| MLD [6] | **0.404** | 0.734 | 10.80 |
| T2M-GPT [69] | 0.514 | **0.745** | 10.92 |
| MotionGPT [25] | 0.510 | 0.680 | 10.35 |
| MDM [55] | 0.497 | 0.396 | 10.84 |
| PriorMDM† [51] | 0.830 | 0.397 | 10.54 |
| GMD† [27] | 1.537 | 0.385 | 9.78 |
| OmniControl [65] | 0.702 | 0.397 | 10.93 |
| Our InterControl | 0.580 | 0.397 | 10.88 |

| Method | Joint | FID ↓ | R-precision ↑ (Top-3) | Diversity → | Foot skating ratio ↓ | Traj. err. ↓ (50 cm) | Loc. err. ↓ (50 cm) | Avg. err. ↓ (m) |
|---|---|---|---|---|---|---|---|---|
| Ours (all) | Root | 0.184 | 0.672 | 9.315 | 0.1044 | 0.0317 | 0.0018 | 0.0693 |
| Ours (all) | Left foot | 0.242 | 0.664 | 9.184 | 0.1005 | 0.0696 | 0.0024 | 0.0671 |
| Ours (all) | Right foot | 0.236 | 0.669 | 9.201 | 0.0983 | 0.0798 | 0.0029 | 0.0680 |
| Ours (all) | Head | 0.172 | 0.678 | 9.359 | 0.0958 | 0.0523 | 0.0044 | 0.0846 |
| Ours (all) | Left wrist | 0.260 | 0.660 | 8.965 | 0.0915 | 0.0375 | 0.0012 | 0.0874 |
| Ours (all) | Right wrist | 0.284 | 0.655 | 9.003 | 0.0920 | 0.0364 | 0.0010 | 0.0872 |

Table 6: **Spatial control** results on the HumanML3D [14] dataset. *Ours (all)* means the model is trained on one randomly selected joint among all joints in each iteration.

Note that methods in the upper part of both tables are unable to perform spatial control, thus they are incapable of generating controllable motions and interactions even if they have lower FID or higher R-precision. For instance, T2M-GPT [69] and MotionGPT [25] tokenize human poses to discrete tokens and is unable to incorporate any spatial information. MLD [6] uses latent diffusion to accelerate denoising steps, yet performing spatial control needs to convert each step of latent feature back to motion representations. It leads to much more computation than MDM [55] and is opposite to MLD's motivation of latent diffusion. Among methods that are suitable for spatial control [51, 27] in Tab. 5, InterControl achieves the best performance in most of semantic-level metrics, and is better than the recent work OmniControl [65] that focuses on single-person motion yet shares similar design of spatial controlling with us.

## B.3    More Single-joint Control Results

In Tab. 1 of our main paper, we have shown the spatial control results with root joint and randomly selected one/two/three joints. Following the recent work [65], we also show the spatial control performance on specific joints in Tab. 6. We find that feet and hands are more difficult to control due to their flexibility, while root (pelvis) and head are more easier to follow, leading to better FID and R-precision.

## B.4    Details of User Study

In the user study, our method generates 50 samples from the contact plans collected from LLM-planner. We also use the original interaction description to generate two-person interactions from ComMDM in PriorMDM [51]. In Fig. 6, we show our designed questionnaire's evaluation instructions and the first question as an example. Each questionnaire has 19 single choice questions randomly sampled from all samples. In the folder named 'user-study-videos', we provide 25 videos sampled from our Intercontrol and PriorMDM for reference.

## Evaluation of two–person motion interactions

This is a set of single–choice questions that will take approximately 5–6 minutes to complete. The questionnaire measures which set of interactions in paired motions better matches the language description and is more natural. The videos show skeleton representations similar to stick figures, with purple and yellow representing two individuals, and each video is about 5 seconds long. The selection priority is as follows: first, choose the interactive motion that more closely matches the language description; if they are equally matched, then choose the motion that is more natural. Naturalness includes whether the movement speed of the person's limbs is reasonable, whether the feet are suspended in the air, whether there is sliding between the feet and the ground, etc. Unreasonable parts are considered unnatural. The descriptions may include words such as swords, shields, mobile phones, and other items, but all the videos only show human skeletons, and the items are not displayed. You can consider whether the set of actions matches the description and is natural if the person is assumed to be holding these items in their hands.

---

*1. Which of the following sets of interactive motions between the stick figures appears more consistent with the description and more natural? Description: One person extends the right hand toward the other.

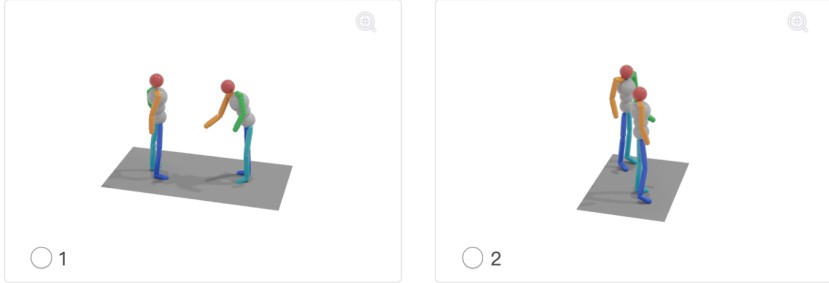

○ 1                    ○ 2

Figure 6: **Example of the questionnaire of user-study.**

## B.5  Details of Evaluation Metrics

Here we select some descriptions for metrics used to evaluate controllable motion generation methods from HumanML3D [14] and GMD [27] to save reader's time.

**Semantic-level Evaluation Metrics from HumanML3D [14]:** Frechet Inception Distance (FID), diversity and multi-modality. For quantitative evaluation, a motion feature extractor and text feature extractor is trained under contrastive loss to produce geometrically close feature vectors for matched text-motion pairs, and vice versa. Further explanations of aforementioned metrics as well as the specific textual and motion feature extractor are relegated to the supplementary file due to space limit. In addition, the R-precision and MultiModal distance are proposed in this work as complementary metrics, as follows. Consider R-precision: for each generated motion, its ground-truth text description and 31 randomly selected mismatched descriptions from the test set form a description pool. This is followed by calculating and ranking the Euclidean distances between the motion feature and the text feature of each description in the pool. We then count the average accuracy at top-1, top-2 and top-3 places. The ground truth entry falling into the top-k candidates is treated as successful retrieval, otherwise it fails. Meanwhile, MultiModal distance is computed as the average Euclidean distance between the motion feature of each generated motion and the text feature of its corresponding description in test set.

**Spatial-level Evaluation Metrics from GMD [27]:** We use Trajectory diversity, Trajectory error, Location error, and Average error of keyframe locations. Trajectory diversity measures the root mean square distance of each location of each motion step from the average location of that motion step across multiple samples with the same settings. Trajectory error is the ratio of unsuccessful trajectories, defined as those with any keyframe location error exceeding a threshold. Location error

is the ratio of keyframe locations that are not reached within a threshold distance. Average error measures the mean distance between the generated motion locations and the keyframe locations measured at the keyframe motion steps.

Table 7: **Detailed prompting example of the LLM Planner.**

| Input |
| --- |

Instruction: two people greet each other with a handshake, while holding their cards in the left hand.

Given the instruction, generate 10 task plans according to the following background information, rules, and examples. Each task plan should completely reflect an entire process of actions described in the instruction.

[start of background Information [

Human has JOINTS: ['pelvis', 'left_hip', 'right_hip', 'left_knee', 'right_knee', 'left_ankle', 'right_ankle', 'left_foot', 'right_foot', 'neck', 'left_collar', 'right_collar', 'head', 'left_shoulder', 'right_shoulder', 'left_elbow', 'right_elbow', 'left_wrist', 'right_wrist' [.

The total number of TIME-STEPS of human motion is 99, the frame-per-second of motion is 20.

The provided text instruction is describing two people performing some actions containing human joint contacts.

The height of all people is 1.8 meters, the arm length is 0.6 meters, and the leg length is 0.9 meters.

Two people are 2 meters away at the beginning (i.e., TIME-STEPS=0).

[end of background Information]

[start of rules]

1. Each task plan should be composite into detailed steps.

2. Each step should contain meaningful joint-joint pairs.

3. Each joint-joint pair should be formatted into {JOINT, JOINT, TIME-STEP, TIME-STEP, CONTACT TYPE, DISTANCE}. JOINT should be replaced by JOINT in the background information. IMPORTANT: The first JOINT belongs to person 1, and the second JOINT belongs to person 2. Each joint-joint pair represents a contact of a joint of person 1 and a joint of person 2. The first TIME-STEP is the start frame number of contact, and the second TIME-STEP is the end frame number of contact. CONTACT TYPE should be selected from {contact, avoid}, DISTANCE should be a float number representing how many meters should be the distance of two joints in the joint-joint pair. For [CONTACT TYPE: contact], the distance of two joints should be SMALLER than the DISTANCE; for [CONTACT TYPE: avoid], the distance of two joints should be LARGER than the DISTANCE. IMPORTANT: Consider the transition of contact types, leave time-steps more than 20 frames without any joint-joint pair between different contact types. Use small DISTANCE variance between different contact types: for the joint-joint pairs that are with [CONTACT TYPE: contact], do NOT use DISTANCE larger than 0.5m in the following [CONTACT TYPE: avoid]; for the joint-joint pairs that are with [CONTACT TYPE: contact], use [CONTACT TYPE: avoid] after 20 frames; for the joint-joint pairs that are with [CONTACT TYPE: avoid], use NO joint pairs for 20 frames if the following CONTACT TYPE is contact. Try to not over-use [CONTACT TYPE: avoid]: if there is no explicit semantics of being far away, just do not use joint-joint pair in that frames; if there is explicit semantics of being far away, then use joint-joint pair with [CONTACT TYPE: avoid].

4. Consider which JOINT will be interacted when two people perform the action described in the text instruction. Translate the text instruction to be steps of joint-joint pairs. Do not include extra joint-joint pairs that is unrelated to the text instruction. IMPORTANT: make joint-joint pairs in different task plans diverse in TIME-STEPS and JOINTs. Each joint-joint contact pairs should be lasting from 3 to 10 frames.

5. Be plausible. Do not generate uncommon interactions. Generate plausible interaction time-steps, and consider the velocity of human motions.

6. Use one sentence to describe what action should person 1 do and one sentence to describe what action should person 2 do according to the text instruction at the beginning of the task plan. IMPORTANT: the sentence starts from 'text 1:' describing the action of person 1 from the perspective of person 1 and the sentence starts from 'text 2:' describing the action of person 2 from the perspective of person 2. Sentences should NOT contain words like 'person 1' or 'person 2', use 'a person' to refer to himself in the sentence and 'others' to refer to others.

7. The steps in the task plan are for both two people. Use one set of steps to describe both two people. The first JOINT belongs to person 1, and the second JOINT belongs to person 2.

8. IMPORTANT: Do NOT add explanations for the steps in task plans. Each step only have one joint-joint pairs.

[end of rules]

[start of an example]

Instruction: two people greet each other with a handshake, while holding their cards in the left hand.

[Start of Plan 1]

Text 1: a person make a handshake with others using his right wrist, while holding his cards in the left wrist.

Text 2: a person make a handshake with others using his right wrist, while holding his cards in the left wrist.

Step 1: {right wrist, right wrist, 0, 10, avoid, 0.3}

Step 2: {right wrist, right wrist, 50, 60, contact, 0.05}

Step 3: {right wrist, right wrist, 90, 100, avoid, 0.3}

[End of Plan 1]

[end of an example]

Table 8: **Example of the LLM generated task plans.**

| LLM Generated Task Plans (only show 5 due to page limit) |
| --- |
| Instructions: The first fencer lunges at the second, who parries the attack and counters with a successful strike to the head. |

[Start of Plan 1]
Text 1: A person lunges towards another with his right foot.
Text 2: A person parries the lunged attack while preparing to counter.
Step 1: {right_foot, left_knee, 5, 10, contact, 0.3}
Step 2: {right_wrist, left_collar, 20, 30, avoid, 0.3}
Step 3: {left_elbow, head, 70, 80, contact, 0.05}
[End of Plan 1]

[Start of Plan 2]
Text 1: A person lunges at the other person with his right foot.
Text 2: A person blocks the lunged attack.
Step 1: {right_foot, left_ankle, 3, 10, contact, 0.2}
Step 2: {right_wrist, right_collar, 20, 30, avoid, 0.25}
Step 3: {left_wrist, head, 70, 79, contact, 0.02}
[End of Plan 2]

[Start of Plan 3]
Text 1: A person takes a lunge step towards another.
Text 2: A person parries the attack and counters.
Step 1: {right_foot, right_knee, 7, 14, contact, 0.3}
Step 2: {left_wrist, right_collar, 22, 30, avoid, 0.25}
Step 3: {right_wrist, head, 69, 77, contact, 0.03}
[End of Plan 3]

[Start of Plan 4]
Text 1: A person lunges northerly towards another with his left foot.
Text 2: A person parries the attack and prepares a counterattack.
Step 1: {left_foot, right_ankle, 6, 10, contact, 0.35}
Step 2: {left_wrist, left_collar, 22, 30, avoid, 0.28}
Step 3: {right_elbow, head, 71, 80, contact, 0.05}
[End of Plan 4]

[Start of Plan 5]
Text 1: A person lunges at another using his right foot.
Text 2: A person deflects the approaching lunge and immediately counters.
Step 1: {right_foot, left_knee, 4, 12, contact, 0.31}
Step 2: {left_wrist, right_shoulder, 20, 30, avoid, 0.3}
Step 3: {right_wrist, head, 73, 81, contact, 0.05}
[End of Plan 5]

Table 9: **Example of processed json file from task plans generated by LLM.**

| Processed Json format, (only show 3 due to page limit) |
| --- |
| Format of 'steps': [ [index of contact joint of person 1, index of contact joint of person 2, start frame, end frame, contact type (contact = 1, avoidance = 0), desired distance (unit as meter) ], ..., ]. |

```
[
    {
        "text_person1": "A person lunges towards another with
his right foot.",
        "text_person2": "A person parries the lunged attack
while preparing to counter.",
        "steps": [
            [11, 4, 5, 10, 1, 0.3],
            [21, 13, 30, 40, 0, 0.3],
            [18, 15, 70, 80, 1, 0.05]
        ]
    },
    {
        "text_person1": "A person lunges at the other person
with his right foot.",
        "text_person2": "A person blocks the lunged attack.",
        "steps": [
            [11, 7, 3, 10, 1, 0.2],
            [21, 14, 30, 40, 0, 0.25],
            [20, 15, 70, 79, 1, 0.02]
        ]
    },
    {
        "text_person1": "A person takes a lunge step towards
another.",
        "text_person2": "A person parries the attack and coun-
ters.",
        "steps": [
            [11, 5, 7, 14, 1, 0.3],
            [20, 14, 34, 42, 0, 0.25],
            [21, 15, 69, 77, 1, 0.03]
        ]
    }
]
```

