# OpenReview forum: "InterControl: Zero-shot Human Interaction Generation by Controlling Every Joint"
_NeurIPS.cc/2024/Conference — NeurIPS 2024 poster_

### Official Review · Reviewer_BPk3 · 2024-07-08

**Soundness:** 3
**Presentation:** 2
**Contribution:** 3
**Rating:** 6
**Confidence:** 5

**Summary:**

This paper focuses on text-to-motion synthesis, specifically, to generate motion of multiple interacting people. The method can potentially work with an arbitrary number of people and models inter-person interactions as pairs of joints that can be either in contact or not, separated by a certain control distance. The method is composed of two main stages: a LLM planer and a diffusion model based on Human Motion Diffusion Model (MDM) that incorporates a motion controller, called Motion ControlNet, and an inverse kinematics (IK) module to have finer control over the joints of generated motion sequences and, thus, generate interactions between synthesized characters. The authors show that the distance between pairs of joints for interactions can be generated with off-the-shelf LLMs. A key feature of this work is that it is zero-shot, thus, the models used do not need to be trained with multi-person motion datasets. However, as it uses a motion controller (based on the ControlNet paradigm) which is fine-tuned to generate global pose sequences (global motion) as opposed to relative motion typically generated by SoTA methods such as MDM. This would be the first method to enable a single-person text-conditioned motion generation for generating interactions for multiple people.

The paper includes comparisons with the SoTA using standard metrics (FID, R-precision, Diversity, Foot Skating, Trajectory error, Location error and Average error). Authors compare with single-person motion generation methods (MDM, GMD, OmniControl). For the two-person setting the paper includes a comparison of Spatial Errors and a user study comparing with PriorMDM.

**Strengths:**

* In part, the paper is clear and well-written.
* The proposed InterControl framework is meaningful.
* Although comparisons with multi-person motion generation models is not comprehensive mostly due to differences in training datasets used (which is understandable), the paper clearly shows an improvement compared to the priorMDM model, both quantitatively and qualitatively.

**Weaknesses:**

### **Method section needs a bit of re-writing**
The method section is somewhat confusing especially in the beginning. There are some definitions that the reader needs to make assumptions about in order to continue reading. For example:
- There is no overview of the method in this section and even if Fig. 2 gives an overview, there is no link to it in the text. I would advise the authors to add this after the formulation and preliminaries so that the reader can easily understand the overall architecture.
- L154: It is not so obvious to me what this discrepancy between relative and global motion is precisely and why it makes priorMDM less controllable in global space. I understand that generating global motion poses different challenges than relative motion, but could the authors explain this further? Doesn't it boil down to generating the global trajectory and then coupling/refining the global motion so that it is coherent with this trajectory as they do in [1]?
- L157: While it may be considered as concurrent work, I suggest the authors discuss a bit [1] which also addresses this issue.
- L174: At this point it is not clear to me what exactly are the spatial conditions. Authors could clarify with a simple example.
- L175+: At this point, it is not clear why the control signal c is potentially sparse? Is it because it comes from text? This is hard to understand as there is no overview pointing out that the method uses a “Contact Plan” based on a LLM.
- L177: I don't understand why the control will be desired for only a select few frames. Don't we want this control all the time? But again, authors may need to first clarify what exactly is this control signal and where does it come from.

### **Additional baselines**
I believe that authors did not include other multi-person motion generation baselines because those explicitly learn from multi-person data. However, I am curious to know how different the results from the InterHuman method differ from this approach?

References:
[1] Zhang, Siwei, et. al, RoHM: Robust Human Motion Reconstruction via Diffusion (CVPR 2024)

**Questions:**

Aside from the questions posted in the Weakness section, I have the following:

### **Need for clarifications**
- I don't fully understand how the model generates the motion for all people. Is it done first individually and then merged somehow? Does the method need to run one MDM+Motion ControlNet instance per person?
- From the qualitative results, it seems to me that I see the effect of an optimization post-processing step after the generation? Do the authors use a post-processing step with the IK or is this the effect of the guidance by the IK module?
- Is the IK module guidance applied both at the end of each denoising step and also after $x_0$ is obtained from the diffusion model?
- To apply the IK module guidance, does the method use the gradient information from the optimization steps to influence or does it apply directly on the estimated poses in a similar fashion as in PhysDiff?


### **Additional comments**

I would suggest including a discussion on the Inter-X dataset at the first paragraph of the introduction. Even though this work is concurrent and this is not strictly necessary, this can make the paper stronger and more up-to-date.

- L34: Could the authors expand a bit about which methods ignore a good design?
- L38: Could the authors include a counter-example for such interactions that do not require 'additional interaction data'. Are these interactions closer interactions such as hugging or, the opposite, more subtle interactions such as talking or holding a meeting?
- L108: I would suggest adding a comment about the method proposed by InterHuman which is also based on the MDM architecture.

### Minor (typos and grammar)
- L155: "is not able" should be “are” instead of “is”.
- L160: is proposed--> proposes
- L166: is “almost” a random noise to be sampled. Almost? Why almost? $x_T$ should be an isotropic Gaussian at T.
- L187: desperately needs a comma, otherwise the sentence completely changes meaning!: "(FK) to convert" -->"(FK), to convert"

**Limitations:**

Yes, the authors include limitations.

---

> ### Author Rebuttal · Authors · 2024-08-07
>
> Thanks for the constructive review. We will revise our paper according to the insightful suggestions in our final version. We are happy that our paper is regarded as ‘clear’ and ‘well-written’ and our framework is ‘meaningful’. We have tried our best to clarify questions and we hope that our response could address your concerns.
>
> **Q1: Paper Writing.**
>
> **A1:** Thanks for your suggestions, we will add the reference to Fig.2 in the beginning of Sec.3 and give more detailed overview of our method.
>
> **Q2: Discrepancy between relative and global motion.**
>
> **A2:** Yes, the discrepancy between relative and global motion is similar to RoHM, where the root joint is represented in global space, while other joints' position are represented in a relative position by setting the origin to the root joint, i.e., relative (local) space. Such data format is adopted in HumanML3D and it is proved to be useful in motion generation by MDM, PriorMDM, etc. However, the global positions of joint in such data format is not available except for the root joint, which makes PriorMDM unable to control joints in the global space. We achieve this by converting such data format to global positions for the input of Motion ControlNet and IK guidance. In RoHM, they also use diffusion models to model root trajectories in global space and other joints' positions in the body-root space, which shares similar idea with motion generation methods in HumanML3D.
>
> **Q3: A concurrent work RoHM for pose estimation.**
>
> **A3:** Thanks for your kind suggestion. We will include this paper in the discussion in the final version.
>
> **Q4: Clarification of spatial condition.**
>
> **A4:** Spatial conditions are coordinate points in the global space, such as the origin (0,0,0). Our framework could accept multiple points together as the spatial condition, e.g., let the left wrist to (1,0,0) and right wrist to (2,0,0) in the same time.
>
> **Q5: Sparsity of spatial condition.**
>
> **A5:** A motion of single person with N frames will have (N, 22, 3) in global space in HumanML3D, where 22 is number of joints and 3 is xyz space. However, for example, we may only set 2 points as spatial condition according to the contact plan among total 22*N joint positions. We want to generate coherent motions with dim (N, 22, 3) by using spatial conditions such as 2 global points. In such cases, spatial conditions are sparse compared to the entire motion sequence.
>
> **Q6: The reason of control being desired for only a select few frames.**
>
> **A6:** For example, when we generate hand-shaking interactions, the contact plan outputted by LLM could only have contact pairs of one's right wrist and the other's right wrist for only 1s (e.g., 30 frames), and does not impose constraints over other frames. In other frames without control, we want such motions to be coherent to the frames being controlled.
>
> **Q7: Qualitative comparison with InterGen.**
>
> **A7:** As our method does not explicitly learn interaction data distributions, it is not suitable to evaluate metrics like FID and Top-3 precision. We shows some qualitative comparison with InterGen in Q1 of general response and Figure 1 of the attached PDF: Our method performs better in fine-grained distances in interactions, e.g.,  it is hard for InterGen to actually let two people holding hands (e.g., distance = 0). The hand-holding interaction in InterGen always leaves some small distances between hands.
>
> **Q8: How the model generates the motion for all people in the inference process.**
>
> **A8:** We generate all people in a batch, e.g., use a tensor of (B, N, D) to represent all people to be interacted, where B is the batch size (number of people), N is frame number, and D is the motion data dimension. The IK guidance is conducted between motions within a batch, leading to IK optimization back-propagated to each individual motion at the same time. Thus, our network is utilized once per denoising step for interaction generation.
>
> **Q9: Post-processing step**
>
> **A9:** We does not use post-processing step. Our IK module is applied at each denoising step in the diffusion process, not at the end of inference process.
>
> **Q10: Details of IK guidance.**
>
> **A10:** No. IK guidance is only applied at the end of each denosing step, like a classifier-guidance operation.
>
> **Q11: Optimization steps in IK guidance.**
>
> **A11:** We use L-BFGS in IK guidance to optimize global joint positions. Like PhysDiff, we don't use this gradient to train network parameters. Instead, the updated motion data feeds into the next denoising step. IK guidance addresses the discrepancy between relative joint data and the need for global position optimization in each step.
>
> **Q12: Inter-X dataset.**
>
> **A12:** Thanks for your suggestion. We will include this paper in our final version.
>
> **Q13: More clarifications.**
>
> **A13:**
>
> L34:
>
> Previous methods could only generate interactions of fixed number of people. Thus, we think they are not 'good' general interaction modeling methods, as they are not flexible to generate group motion with arbitrary number of people.
>
> L38:
>
> Thanks for this question. We think the counter-example is more like 'hugging', where close interactions are difficult to be described only by distance between joint pairs: it also require the mesh penetration control to avoid unnatural interactions with penetrations. On the contrary, for talking and holding a meeting, we could use a constant distance to constrain two people's root joints (e.g., 2 meters) and also constrain their head orientation to be face-to-face to make such interaction looks realistic.
>
> L108:
>
> Thanks for the suggestion. We will include the discussion of InterGen in our final version.
>
> Typos:
>
> Thanks for pointing out the typos. We will carefully revise our paper.
>
> L166: Yes, x_T is a random gaussian noise;
>
> L187: Yes, we should add a comma after (FK).

---

> ### Comment · Reviewer_BPk3 · 2024-08-12
> **Feedback**
>
> I thank the authors for their responses and the rebuttal document. I have some additional comments/questions.
>
> #### **Comparison with OmniControl.**
> In Table 1, the authors do not specifically state how they deal with the randomly selected joints for comparison. Are these joints the same for both OmniControl and the proposed method for each of the text prompts used? Or are these joints randomly selected for each generation independently?
>
> Also, as pointed out by another reviewer, directly comparing the controllability of the method with other single-person methods (e.g., OmniControl) can make the paper stronger and more convincing.  Furthermore, it seems that OmniControl’s code was already published around mid December 2023, so I would dare to say that considering this work concurrently is walking on a thin line.
>
> #### **Qualitative results.**
> I agree with other reviewers that the qualitative results present abrupt changes in velocity, especially when contact between two people is enforced. It seems that the guidance of the optimization step is “snapping” two joints together. This may show that the guidance step has an overly important role on the final results which also makes the generated motions look less natural than current methods.
>
> #### **Clarification of method’s details.**
> L225-228. Consecutive sentences between these lines seem to directly contradict each other. Here it is stated that IK guidance is applied when training the ControlNet, but then it states that IK guidance eliminates the need for training Motion ControlNet. This does not make sense to me.
>
> #### **Correction about guidance.**
> Based on the author’s response to the questions related with IK guidance, then I would say that the IK guidance is NOT like classifier-free guidance as it does not use the gradients to guide the generation. As in PhysDiff, this method uses a direct modification of the pose during the denoising process. Thus, this is indeed a type of guidance but it is not like classifier guidance or done in “a classifier guidance [9] manner,”  (L204-205). I suggest authors change the wording in the paper to make this clearer.
>
> #### **Writing.**
> L199: Using e.g. to refer to the single person motion dataset used for training makes the wording confusing. Are the ControlNet authors present in this work trained with HumanML3D only or is it trained with more datasets similar to this one?
>
> Having said all of these. I am still a bit concerned that the method section is not clear at this current stage and may need important re-writing and will slightly lower my rating.

---

> > ### Author Response · Authors · 2024-08-12
> >
> > Thanks for your feedback!
> > ### Comparison with OmniControl
> > The 'random one/two/three' item in Table 1 means one/two/three joints randomly selected for each generation independently. OmniControl also adopts similar strategy in the inference process. Yet it is hard to ensure that 'joints the same for both OmniControl and the proposed method for each of the text prompts used', as the joints are randomly sampled for each text prompt and OmniControl does not provide such configuration file to record the randomly sampled joints. For controlling specific joints, please refer to Table 6.
> >
> > For the concurrent work statement, our paper was done in mid November 2023 and then submitted to the CVPR 2024 conference (We could provide such submission record to ACs and Reviewers if needed). Our paper's code is independently developed and we made it available to CVPR reviewers in late November 2023. Please consider this information to compare our methods with OmniControl.
> >
> > ### Qualitative results
> > Thanks for watching our qualitative results. As we mentioned in the general response of rebuttal, our qualitative results provided in supp. mat. were not processed by 1d gaussian filter. Yet, we find later that many methods commonly adopt it to promote the smoothness of their qualitative results, such as InterGen. In Q4 of the general response of our rebuttal, we have shown that our motion shows similar acceleration with InterGen with the same post-process with InterGen.
> >
> > ### L225-228
> > Thanks for the detailed review. IK guidance could be used in two types of intermediate result: $\mu$ or $x_0$. IK guidance is applied when training the ControlNet if we use $\mu$. IK guidance eliminates the need for training Motion ControlNet if we use $x_0$. The detailed algorithm could be found in Algorithm 1 of our Appendix, which uses  $\mu$.
> >
> > ### Correction about guidance
> > We agree that  IK guidance is NOT like classifier guidance as it does not use the gradients to guide the generation. Yet it is like classifier guidance in some degree to guide the denoised results in each denoising step, which is certainly dissimilar to classifier-free guidance. Besides, OmniControl uses the standard classifier guidance. We will revise such description of IK guidance to make it more precise, such as 'IK guidance utilizes IK to guide the pose during the denoising process similar to PhysDiff'.
> >
> > ### L199
> > We use 'e.g.' to say that our model could also be trained on other single-person data. In our interaction experiments, all results are generated by model trained on HumanML3D. Yet, it could be further improved by training on larger single-person datasets. We will revise such description in the final version to make it precise.

---

> > > ### Comment · Reviewer_BPk3 · 2024-08-12
> > > **L225-228**
> > >
> > > The final model/architecture authors present in the paper, makes use of $\mu$? The algorithm mentioned, as stated here uses $\mu$, thus, the final model does that also, is this correct?

---

> > > > ### Author Response · Authors · 2024-08-12
> > > >
> > > > Yes, we quantitatively find that $\mu$ leads to a little better performance than $x_0$ in our ablation study: item (5) in Table 4 of Appendix shows performance using $x_0$, while item (1) shows performance using $\mu$. Therefore, our final model adopts $\mu$ in IK-guidance, which requires the IK-guidance in training Motion ControlNet. We would really appreciate your consideration  in raising the final rating if our explanation addresses your concerns.

---

### Official Review · Reviewer_7GL6 · 2024-07-12

**Soundness:** 3
**Presentation:** 3
**Contribution:** 3
**Rating:** 5
**Confidence:** 4

**Summary:**

This paper tackles the challenge of generating human interaction motions involving a flexible number of characters. To simplify the representation of these interactions, the authors propose a joint-pair contact/avoid representation. Given an interaction description, a large language model (LLM) generates motion instructions and identifies contact joint pairs. These outputs serve as inputs for the subsequent motion diffusion model. The Motion ControlNet is trained to produce motion sequences based on the motion instructions and to optimize the joints to align with the specified joint contact pairs.

**Strengths:**

1. This paper is addressing an important human interactions learning task that is not well studied due to the lack of multi-person interactions datasets, and this paper suggests a framework that only leverages the single-person dataset to produce reasonable multi-person interactions.
2. The joint-pair distance representation of human interactions are simple and the IK-guidance shows to be effective.
3.  The manuscript is well-written and easy to follow.

**Weaknesses:**

1. It is not clear how does the contact joint pairs for each frame are generated by LLM. A joint jump usually presents when the contact between two humans happens, and what can be this unsmooth motions artifacts result from?
2. I still observe that there are still severe human collision happens in some generated human interactions. The joint pair contact type produced from LLM might not be fine-grained enough to produce natural and plausible interactions. The followed physical simulator tracking step might mitigate the collision artifacts a bit, but still it needs better spatial control over body roots to avoid severe collisions during motion.
3. The iterative IK guidance based optimization happens at each diffusion step, and this might result in less efficient framework. Might directly optimise the noise like done in DNO [1] or a single-step post-optimisation also work?

[1] Optimizing Diffusion Noise Can Serve As Universal Motion Priors, CVPR 2024

**Questions:**

This paper is addressing an important task - human interaction generation, by only leveraging single-person dataset. In addition to some of my questions presented in the weakness section, I have a few general questions here, and I am glad to hear some feedback or insights from authors:
1. If we want to scale up to more involved human characters
* How about the efficiency at inference time of the proposed pipeline?
* Would LLM planning still be effective and fine-grained enough for multiple-person interactions?
2. It would also be insightful to see the comparison with Intergen or fine-tuning on InterHuman dataset as well in the two-person interaction scenario.

**Limitations:**

Yes, the authors have addressed the limitations.

---

> ### Author Rebuttal · Authors · 2024-08-07
>
> Thanks for the insightful review. We will revise our paper according to the constructive suggestions in our final version. Please refer to General Response for comparison with OmniControl and InterGen, and the explanation of penetration and unsmooth motion issues.
>
> **Q1: How does the contact joint pairs for each frame are generated by LLM?**
>
> **A1:** The format of contact joint pairs (contact plans) is in Table 9 of our Appendix. The prompt for generating contact plans by LLM is shown in Table 7 of our Appendix, where the raw output of LLM is in Table 8.
>
> **Q2: Comparison with DNO, and efficiency issues.**
>
> **A2:** Thanks for the insightful suggestion. Firstly, the concurrent work DNO is a good exploration of motion editing which shares similar idea with our IK guidance. However, it also require 300 or 500 steps for motion editing or refinement, which takes more than 3 minutes according to their paper (Sec.6 in page 6). On the contrary, our method need 80s for inference, while previous methods takes more seconds to control the motions following spatial conditions (GMD needs 110s and OmniControl needs 120s). Secondly, if the inference speed is really important, we could use other speed-up techniques in diffusion models to improve the inference speed, such as DDIM and Consistency Model. For example, our framework could utilizes the recent MotionLCM (https://dai-wenxun.github.io/MotionLCM-page/) to speed up the inference to 30ms per denoising step and get realistic motions within 4 steps. Yet, our main contribution of zero-shot interaction generation ability will not be influenced.
>
> **Q3: Scale up to more characters.**
>
> **A3:** Our framework is able to perform motion generation in a batch to speed up multi-human motion generation, where all characters' motion are generated together. Thanks to the batch computation ability in GPUs, the inference time is almost the same with single-person motion generation. The IK guidance optimization process will add little extra burden with the increase of number of characters. In practice, 2-character inference takes about 80s and 3-character inference takes about 90s with a standard 1000-step DDPM inference. As we mentioned above, we could use MotionLCM to further speed up the inference process of multi-character interaction generation.
>
> **Q4: LLM planning for more characters**
>
> **A4:** We do not collect contact plans of more characters from LLM for quantitative experiments. Yet, as we show in the supp. mat., LLM works well in the three people cases of fighting and holding hands by providing meaningful prompts. Furthermore, we believe the ability of LLM will be better in the future to handle more complex interactions. Our current experiments illustrate that GPT-4 works well in the two-people interaction contact plan generation. And it also works in some three-people interaction cases. Finally, LLM is a tool to scale up the contact plans in a batch. Yet, our framework does not necessarily need a LLM to produce reliable contact plans: contact plans could also be provided by professionals such as artists. Instead of manually designing keyframes, writing contact plans could greatly alleviate their efforts to generate interacted motions in their work flow.

---

> > ### Comment · Reviewer_7GL6 · 2024-08-12
> >
> > I appreciate a lot for the detailed response from authors, and I have read the rest reviews and authors' responses as well. Many of my questions are addressed by authors. Overall, the proposed zero-shot human interaction generation pipeline with LLM-based contact planning together with IK-guided optimization shows to be effective, though it is still challenging to generate very realistic and plausible interaction motions. Also, I agree with the other two reviewers that it is encouraged that the author includes further clarifications on the differences and comparisons between the intercontrol and intergen and omnicontrol. Additionally, what would further complete this work is to compare the proposed intercontrol with data-based methods trained on multi-person dataset, and this would be insightful for the community. I would like to keep my original score as borderline accept after reading authors' response and other reviews.

---

> > > ### Author Response · Authors · 2024-08-12
> > >
> > > Thanks for your kind feedback. We sincerely appreciate your effort to review our paper.

---

### Official Review · Reviewer_HwRy · 2024-07-13

**Soundness:** 3
**Presentation:** 3
**Contribution:** 2
**Rating:** 5
**Confidence:** 4

**Summary:**

The paper introduces InterControl, a method designed to address the task of controllable human motion generation and zero-shot human interaction generation. By leveraging the prior knowledge of LLM, InterControl can generate human interactions involving any number of people in a zero-shot manner. Specifically, the authors utilize LLMs to convert textual descriptions of interactions into contact plans, transforming the task of multi-person interaction generation into single-person motion generation. The InterControl model, which is based on Motion ControlNet and IK Guidance, is then used to achieve controllable single-person motion generation. Experimental results demonstrate the superiority of InterControl over previous controllable motion generation methods and its ability to produce realistic human interactions.

**Strengths:**

- The proposed task(zero-shot human interaction generation) is interesting, important, challenging, and meaningful.
- The related work section is great, providing an excellent summary of relevant research.
- The proposed method makes sense and is logically sound.
- The results are reasonable. I particularly appreciate the application section, where the character interactions in the simulation are executed very well.

**Weaknesses:**

- The qualitative results are not really good. a) The supplementary video shows some cases where some joints have abrupt velocity changes, which conflicts with L232: "IK guidance can adaptively modify velocities from the start to frame n." b) There are some cases of body penetration and unrealistic interactions in the three-people-fighting/two-people-fighting scenarios.
- It might be beneficial to incorporate some visualization of the contact plan. According to Sec 3.1, contact should include both contact and avoid forms, described by distance d. However, in the supplementary demo, I only observe d=0 contact. I would like to see some examples of avoidance and how non-zero distance contact plans guide motion generation.
- Maybe more discussion is needed on the differences between omnicontrol and intercontrol, as they are similar methods (both of InterControl/OmniControl use ControlNet-like framework and classifier guidance). The author may want to discuss if using OmniControl as the control yields better results in the task of zero-shot human interaction generation?
- The paper didn’t compare InterControl with data-based methods such as InterGen. The author may want to demonstrate the superiority of InterControl as a zero-shot method, possibly by highlighting interactions that InterGen cannot generate (2 people interaction).
- It might be beneficial to include some qualitative comparisons between InterControl and state-of-the-art methods(e.g. OmniControl) in the task of Single-Person Controllable Motion Generation to better demonstrate the superiority of InterControl.

**Questions:**

- What is the corresponding text for the supplementary material?
- Can the LLM really provide a reliable contact plan?

**Limitations:**

Although the authors included a "Limitations'' subsection in Sec.5, from my perspective, they did not clearly claim the limitations of the method. The authors may want to point out that the realism of the interactions generated by InterControl heavily depends on the LLM's correct interpretation of the text, which cannot be guaranteed.

---

> ### Author Rebuttal · Authors · 2024-08-07
>
> Thanks for the insightful review. We will revise our paper according to the constructive suggestions in our final version. Please refer to General Response for comparison with OmniControl and InterGen, and the explanation of penetration and unsmooth motion issues.
>
> **Q1: Examples of avoidance.**
>
> **A1**: Thanks for the kind suggestion. We do have examples of avoidance in our qualitative results, yet we did not annotate them out. For example, the 11-th second of video 'two-people-dancing.mp4' (or the Figure 1 of the attached PDF in the author rebuttal) shows an example of contact and avoidance at the same time: when two characters are holding hands, their other hands are away from each other by at least 2.4 meters. Such avoidance leads to stretching dance motions, which could not achieved by joint contact alone.
>
> **Q2: Discussion of the differences between omnicontrol and intercontrol.**
>
> **A2**: First of all, we want to clarify that our work is an concurrent work with Omnicontrol, and we are not aware of this paper when we did our work. We have discussed the difference of technical contribution in Sec.A.4. Besides, our work shows notable improvement of non-root joint control over OmniControl in HumanML3D dataset. We have included the zero-shot interaction generation results of OmniControl in the Table of A2 in general response.
>
> **Q3: Qualitative comparisons between InterControl and OmniControl.**
>
> **A3**: We have include detailed quantitative comparsion with OmniControl in Table.1. Besides, our main contribution is the ability of zero-shot interaction generation. The proposed single-person spatial control is an approach to achieve our main contribution. In Figure 2 of the attached PDF in the author rebuttal, we show a qualitative comparison between InterControl and OmniControl, where our method shows better hand joint control in hand-shaking.
>
> **Q4: Corresponding text for supp. mat.**
>
> **A4**: For user study videos, the texts are in 'two_person_text_prompts.txt'. For visualization videos, such texts are writted by ourselves to show in-the-wild results. Here are specific texts: 'two-people-dancing.mp4': Two people are dancing, sometimes they stand in an open dance position: one hand joined together, while their other hands are extended outward. 'two-people-winning-gesture.mp4': Two people walk slowly, their arms raised high while holding each other's hands, displaying a triumphant gesture after a fighting contest. 'three-people-holding-hands.mp4': One person held the hands of two others, each with one hand, forming an arc with all three facing the same direction. 'three-people-fighting.mp4': Two people are fighting against a third person, using their wrists and feet to attack. The third person is also counterattacking.  'two-people-fighting.mp4': One person is fighting against another person, using his wrists and feet to attack. The second person is also counterattacking.
>
> **Q5: Can the LLM really provide a reliable contact plan?**
>
> **A5**: Thanks for this insightful review. We agree that LLM is not guaranteed to provide a reliable contact plan in our framework. As our main motivation and contribution is to design a zero-shot interaction generation framework, we leverage LLM to provide necessary information for this goal. From our empirical results, we think LLM (e.g., GPT-4 in our experiments) could provide a reasonable contact plan in most cases, which is agreed by many previous works in robotics, such as [64] and [a]. Whether or not LLMs could guarantee to generate reliable contact plan is beyond our paper's scope.
>
> **Q6: Limitations.**
>
> **A6**: We agree that the LLM's correct interpretation of interaction descriptions cannot be guaranteed. As our paper's focus is not LLM, our framework provide a pioneer exploration to perform zero-shot interaction generation, and we empirically find that LLM works well in most cases. With the improvement of LLM's knowledge and robustness in the future, we believe the contact plan generated by newer LLMs will be more reliable.
>
> [a] Song, Chan Hee, et al. "Llm-planner: Few-shot grounded planning for embodied agents with large language models."  *in CVPR,* 2023.

---

> > ### Comment · Reviewer_HwRy · 2024-08-14
> >
> > I greatly appreciate the authors' detailed response, which has addressed most of my concerns. In the response, the authors included comparisons between InterControl, InterGen, and OmniControl, provided examples of avoidance, and supplemented the text descriptions in the video. I am grateful for the authors' efforts in addressing and expanding on these issues.
> > Although I suspect that the LLM may not be able to provide a reasonable contact plan "in most cases," and the qualitative results generated by InterControl are not perfect, I think that the novelty of the zero-shot interaction generation task and InterControl outweigh these shortcomings. After reviewing the supplementary materials, considering the other reviewers' opinions, and the authors' response, I am willing to change my score to borderline accept.

---

> > > ### Author Response · Authors · 2024-08-14
> > >
> > > Thank you for your thoughtful reply. We sincerely appreciate your effort in reviewing our paper and considering our rebuttal.

---

### Official Review · Reviewer_VAgJ · 2024-07-13

**Soundness:** 2
**Presentation:** 3
**Contribution:** 2
**Rating:** 4
**Confidence:** 5

**Summary:**

### Summary

The paper "InterControl: Generating Human Motion Interactions by Controlling Every Joint" aims to generate interactions between multiple people based on text descriptions, with precise joint control. It leverages a pre-trained single-person motion diffusion model and extends it to multi-person scenarios using a large language model (LLM) to guide joint interactions.

The methodology involves:
- Using an LLM planner to map out interactions and distances between joints.
- Combining priorMDM with ControlNet, conditioned on LLM-extracted distances.
- Optimizing interactions with LGBFS to enhance performance.

Evaluations are conducted using Text2Motion datasets and user studies, focusing on joint control and positional accuracy. The approach also assesses 100 interactions from the InterHuman dataset.

### Contributions

1. **Dynamic LLM Guidance**: Introduces LLMs to guide joint interactions dynamically, making the process scalable and suitable for large-scale data generation.
2. **Precise Joint Control**: Allows precise control over any joint, improving flexibility compared to methods requiring predefined control signals.
3. **Multi-Person Interaction**: Extends a single-person model to handle multi-person interactions, demonstrating scalability.
4. **Enhanced Visualization**: Suggests using human body models like SMPL or GHUM for better visualization and interpretability of interactions.
5. **Comprehensive Evaluation**: Provides thorough evaluation and comparison with previous models, addressing issues like interpenetration and alignment between control and text prompts.

While the paper presents promising results and tackles the challenging task of generating realistic multi-person interactions, it also identifies areas for further improvement, such as better handling interpenetration and improving alignment between spatial control and text conditions.

**Strengths:**

- The use of large language models (LLMs) to dynamically guide joint interactions introduces a scalable and innovative approach to multi-person motion generation.
- The method allows for precise control over any joint at any time, enhancing flexibility and improving upon previous methods that relied on predefined control signals.
- The approach successfully extends a single-person motion model to generate realistic interactions among multiple people, demonstrating its scalability and applicability to complex scenarios.
- The paper is well-written and easy to follow, with a clear presentation of previous works and a strong introduction that provides good context.
- The simplicity of the IK guidance and LLM-generated contact points helps automate the generation process, which is beneficial for large-scale data generation pipelines.
- The paper includes comprehensive evaluations and comparisons with existing models, showing that the results for single-joint control and multi-person interactions are indeed better in several aspects.
- The integration of ControlNet and LGBFS optimization with LLM planning is a novel contribution, pushing the boundaries of 3D human motion generation.

**Weaknesses:**

- The contributions of the paper seem limited, with similarities to existing works like OmniControl and GMD.
- The generated motions often suffer from interpenetration, which reduces the realism of the interactions.
- There are discrepancies between the spatial control and text conditions, suggesting that the alignment between them needs improvement.
- It is unclear how ControlNet is finetuned, especially regarding the use of the HML3D dataset and the extraction of necessary control features.
- A simpler baseline for generating each person’s motion and optimizing interperson distances is missing, which could highlight the necessity of additional modules.
- The qualitative results lack sufficient visualization, making it difficult to assess the plausibility of interactions from the provided videos.
- The method focuses on multi-person motion generation but only evaluates single-person text-to-motion datasets, missing evaluations on available multi-person datasets.
- The use of 3D joint locations without a human body surface makes it hard to judge motion plausibility and perceive contacts between people.
- The presentation lacks clarity in some sections, particularly in explaining complex steps and figures, such as the Gaussian noise issue and Figure 2.
- The approach relies on LLMs for joint distances, which may hallucinate content and lead to errors in generating plausible contact maps.
- The work does not include suitable multi-person motion capture data during training, limiting the robustness of the interactions modeled.
- The focus on joint-to-joint contacts overlooks other types of human contact, such as grabbing an arm, which are not modeled effectively.

___

Small notes
- The teaser is misleading as it suggests conditioning on images for interaction generation.
- The claim that TEMOS performs worse than MDM is not substantiated; TEMOS actually performs better in some cases.

**Questions:**

### Questions and Suggestions for Improvement
(importance sorted)

**Evaluation and Results**:
   - Expand evaluations to include more comprehensive datasets, especially those involving multi-person interactions, and provide a detailed performance analysis.
   - Provide an analysis of the LLM's error rates in generating plausible contact maps and discuss how this impacts the model's performance.
   - Evaluate the model on available multi-person datasets and collect text descriptions for these datasets to provide a thorough assessment.

**Methodological Improvements**:
   - Address the issue of interpenetration in generated motions by refining IK guidance and adding constraints to improve realism.
   - Elaborate on the process of fine-tuning ControlNet on the HML3D dataset and extracting necessary control features.
   - Implement and compare a simpler baseline that generates each person’s motion using existing methods and optimizes interperson distances using LGBFS.

**Clarification of Contributions**: Clearly distinguish the paper's contributions from existing works like OmniControl [49] and GMD [26]. Highlight the novel aspects and how this work advances the field beyond these prior studies.

**Visualization and Presentation**:
   - Use human body models like SMPL for better visualization of motions, facilitating better perception of contact and interaction.
   - Include mesh-based visualizations and use original AMASS skeletons for training to enhance interpretability and clarity of results.
   - Improve the clarity of the presentation, particularly in explaining complex steps and figures, such as the Gaussian noise issue and Figure 2.

By addressing these points, the paper can provide clearer contributions, ultimately strengthening the overall quality and impact of the work.

**Limitations:**

The Limitations are discussed.

---

> ### Author Rebuttal · Authors · 2024-08-07
>
> Thanks for the detailed review. Please refer to general response for more common questions.
>
> Q1: Similarity to OmniControl and GMD.
>
> A1: (1) Our method could control all joints while GMD only controls root joint. (2) We focus on zero-shot interaction generation and use controllable single-person motion generation as the approach to achieve this goal, while OmniControl only consider single-person motion. Our method shows better joint control ability in non-root joint control (refer to Table 1 of our paper, and qualitative results). Besides, our work is concurrent to OmniControl and we design InterControl independently.
>
> Q2:  Discrepancies between the spatial control and text conditions.
>
> A2: It is unclear the meaning of discrepancies. We will appreciate it if the reviewer could elaborate it.
>
> Q3: How Motion ControlNet is finetuned.
>
> A3: We have included the training details of Motion ControlNet in Line 193-199. The control features are global positions converted from HumanML3D data format, by using forward kinematics.
>
> Q4: A simpler baseline for optimizing inter-person distances on single-person motion should be compared.
>
> A4: It is unclear that how the 'inter-person distances' in the review is defined. If it is defined on the distance of any joints from different people, our method itself is similar to such 'simple baseline'. If is is only defined on root joint, GMD is such baseline for spatial control, which has been included in the spatial control comparison quantitatively.
>
> Q5: Qualitative results lack sufficient visualization.
>
> A5: We have provided 3D-skeleton visualization on user study cases and in-the-wild cases with both two-people and three-people scenarios. Furthermore, we provided physical animation results from our generated kinematics-based motions for better illustration of the execution of a hard interaction case: fighting. The effectiveness of our visualization is agreed by reviewer#HwRy.
>
> Q6: Missing training and evaluations on multi-person datasets.
>
> A6: As our method is a zero-shot interaction generation method, we only train our method on single-person motion datasets and it does not learn data distribution on multi-person datasets. Thus, it is not suitable to evaluate FID or Top-3 precision on these benchmarks. The usage of multi-person motion capture data during training is beyond this paper's scope and we leave this to our future work.
>
> Q7: 3D joint locations without a body surface makes it hard to judge motion plausibility and perceive contacts.
>
> A7: As the mainstream motion datasets adopt 3D joint locations as motion data, it is more direct to visualize motions with 3D skeleton. SMPL visualization used by previous methods need an additional step that utilize SMPLify method to convert 3D joint locations to SMPL meshes. However, such conversion will introduce errors and the converted mesh will not be faithful to the original motion output. Furthermore, the mainstream motion datasets such as HumanML3D commonly remove the hand joints from SMPL, leading to a 22-joint data format instead of 24-joint data format in original SMPL. Therefore, many interactions could only be achieved by using wrists instead of hands in current interaction results. The effectiveness of our 3D joint location visualization is agreed by three other reviewers. Finally, we have shown that physical animation could be useful to improve the plausibility of both motion and contact. Our method has the potential to further be improved by leveraging surface optimization methods.
>
> Q8: More explanations on Gaussian noise and Figure 2.
>
> A8: We will carefully revise our paper in the final version.
>
> Q9: The LLM may hallucinate content and make errors in generating contact maps
>
> A9: Thanks for this suggestion. We agree that LLM could hallucinate contents in our framework. As our main motivation and contribution is to design a zero-shot interaction generation framework, we leverage LLM to provide necessary information for this goal. From our empirical results, we think LLM (e.g., GPT-4 in our experiments) could provide a reasonable contact plan in most cases, which is agreed by many previous works in robotics, such as [64] and [a].
>
> Furthermore, we believe the contact plan generated by LLM will be more reliable with the new improvements of LLMs. Whether or not LLMs could guarantee to generate reliable contact plan without hallucination is beyond our paper's scope.
>
> Q10: The focus on joint-to-joint contacts overlooks other types of human contact, such as grabbing an arm.
>
> A10: Thanks for this insightful suggestion. We agree that our joint-to-joint contacts could not handle all kinds of human contact, such as grabbing or hugging. We have clearly discussed such limitation of interaction definition in Line 35-43. Furthermore, our method could be seamlessly extended to joint-to-bone contacts by sampling keypoints in the bone. It could be easily achieved by interpolation between two adjacent joints.
>
> Q11: Teaser could be misleading.
>
> A11: Thanks for the suggestion. We want to illustrate that our definition of interactions could result in meaningful human interactions in our daily life. As we clearly mentioned zero-shot motion interaction generation in the title and paper, we believe the teaser will not be misleading or hard to understand, which is agreed by other reviewers.
>
> Q12:TEMOS performs worse than MDM is not substantiated.
>
> A12: We does not claim that TEMOS performs worse than MDM in our paper. We will appreciate that if reviewers could check our paper again.
>
> Q13: Analysis of the LLM's error rates.
>
> A13: As LLM is not our focus, we empirically show that LLM could work well in the interaction generation. It is worth noting that we use an off-the-shelf LLM without any finetuning. The effectiveness of LLM has been agreed in many previous works, such as [64] and [a].
>
> [a] Song, Chan Hee, et al. "Llm-planner: Few-shot grounded planning for embodied agents with large language models."  in CVPR, 2023.

---

> > ### Comment · Reviewer_VAgJ · 2024-08-10
> > **Feedback**
> >
> > - ### Comparison with InterGen
> > InterGen is dataset-based method, so it may not be able to generalize to unseen interactions. Is this interaction within its scope and training set? Since you are not training in their dataset, it would be nice to provide some more details in this comparison. Your generalizability is clearly superior, but comparisons should be fair.
> > - ### OmniControl
> > Indeed your method performs slightly better than OmniControl. However, the IK-guidance could results in such better interactions. OmniControl results seems close to yours. Are you visualizing the results before or after IK-LGBFS optimization? What happens if you apply the same optimization to OmniControl?
> > - ### Teaser
> > I have read the rest of the reviews and haven't seen any agreement on the teaser's relevance/appropriateness. Showing images is not so scientific since this is neither your contribution or input to the model. The input are text descriptions and your not guiding your interactions through poses. Hence, this is not even in the scope of this work. It could be a potential application after some retrieval method. Teaser images are normally used as a gist of a method and should avoid overstating things.
> > - ### Ablating LLMs
> > I manually checked your plans and seem quite accurate in general. However, it would be nice if you added some extra analysis on failure cases or some -at least empirical- discussions on it.
> > - ### 3D joint locations without a body surface makes it hard to judge motion plausibility and perceive contacts.
> > I kindly disagree that joint locations is common representation for human motion. The largest database (AMASS), which most of the works benefit from adopts SMPL rotations as a representation which enables using a full body mesh. The adoptions of joint positions and non-SMPL rotations from users of datasets such as HML3D and the post-processing optimization they commonly performs is a technique used by such works for text-to-motion, but there are other works which predict SMPL rotations and meshes directly. TUCH (CVPR 2021) highlights the importance of the human body surface for self, and person-to-person interactions. I would expect retraining MDM with SMPL rotations - done in STMC(https://mathis.petrovich.fr/stmc/) - and using such features to control the generated motions.
> >
> > ### Final Remarks
> >
> > The zero-shot manner that is used for the interactions is nice. The proposed LLM-based planning is clear and seems to work fairly well.
> > - The surface of the human body is the appropriate way for this to be shown and achieved.
> > - Except from qualitative comparisons, is there any quantitative comparisons on the datasets proposed in InterGen?
> > - I suspect that OmniControl which such IK guidance could perform similarly. Is that the case?

---

> > > ### Author Response · Authors · 2024-08-11
> > > **Response to the feedback of reviewer#VAgJ (2/2)**
> > >
> > > ### Response to Final Remarks
> > >
> > > The surface of the human body could be beneficial, but it does not affect our main contribution. We think it will be better to try it as an individual paper in the future work.
> > >
> > > Quantitative comparisons with InterGen will be unfair for our method, as our method is a zero-shot method and InterGen is fully-supervised.
> > >
> > > IK guidance is one of our major difference with OmniControl. Comparing our methods with IK-guidance + OmniControl is like an ablation study of our method, which has been quantitative compared in Table 4 of our Appendix.

---

> ### Author Response · Authors · 2024-08-11
> **Response to the feedback of reviewer#VAgJ (1/2)**
>
> Thanks for your kind feedback! We will address your concerns point by point.
>
> ### Comparison with InterGen
>
> As you mentioned, InterGen is a data-driven method while our method is a zero-shot method. Our definition of interaction and the training dataset is totally different with InterGen. Thus, our method is not comparable with InterGen. For comparable methods like GMD, we have conducted extensive experiments in our paper to compare with them. According to the request of reviewers, we qualitatively compare our generalization ability on some distance-sensitive interaction cases with InterGen, which is in the rebuttal PDF. As we do not train our method on InterHuman, the comparison with InterGen within its data distribution will be unfair for us: it is obvious that zero-shot method is hard to match the performance of fully-supervised methods, especially on the generation tasks that require the model to learn data distribution.
>
> By comparing with InterGen qualitatively on some distance-sensitive interactions, we show that the generalization ability could be an advantage of our method over InterGen. Yet, quantitatively comparing our method with InterGen is an unfair comparison.
>
> ### Comparison with OmniControl
>
> The major difference with OmniControl is (1) our IK-guidance and (2) the non-root joint control ability in Motion ControlNet. As OmniControl adopts a gradient-based optimization method similar to classifier-guidance, it requires more steps (slow inference speed) and lead to worse joint alignment results than ours. Our IK-guidance draws the intuition of IK that is effective and quick for optimization in diffusion models, as the optimization of joint locations is 2nd-order differentiable (unlike the classifier-guidance in image generation). On the contrary, OmniControl directly follows classifier-guidance. If we add IK-guidance to OmniControl, it will be very similar to our method itself. So conducting comparison mentioned by reviewer is like comparing two versions of our own method. Such result has been quantitatively compared on single-person dataset in Table 4 of our Appendix (effectiveness of IK guidance: item 4,5,6 compared with item 1; effectiveness of Motion ControlNet:  item 2,3 compared with item 1).
>
> ### Teaser
>
> As I mentioned ‘agreed by other reviewers’, I mean no other reviewers mention that our teaser is misleading. I agree with reviewer that image is not our contribution/scope. We show images to illustrate that our distance-based interaction definition commonly exists in many scenarios in our daily life and it is effective to represent a large portion of interactions. Such definition could results in meaningful interactions, which could also be found in the Internet images. We will adjust our teaser and consider to remove these images and only keeps interaction visualizations.
>
> ### Ablating LLMs
> Thanks for your effort to check our LLM plans. As you mentioned, our LLM plans seem quite accurate in many simple interactions, such as shaking hands. Here we provide an empirical discussion on failure cases: (1) Very close whole-body interactions, such as hugging. It is difficult to describe joint distances in very close whole-body interactions, even by humans. Yet, such interactions are also hard for data-driven methods, e.g., InterGen also shows artifacts like penetration in hugging on their homepage demos (https://tr3e.github.io/intergen-page/). (2). Distant Interactions, such as playing tennis. The distance between two players is unclear when they play tennis, yet they actually are interacting through the tennis ball. All we can do in this case is to restrict two players in the tennis court by setting their root joint within some region by our IK guidance.
>
> ### 3D joint locations v.s. mesh surfaces
>
> As HML3D is the current largest text-annotated mocap dataset, many previous methods follow its data format and setting, such as MDM, PriorMDM, T2M-GPT, MLD, OmniControl, etc. Besides, InterHuman dataset also adopts 3D joint locations instead of original SMPL rotations (page 7 of intergen paper). We want to emphasize that STMC is not publicly available when we developed our InterControl. Actually our work is done about 2 month earlier than it. I understand that original SMPL representation could be beneficial to modeling human interactions. Yet, I think it is more like a problem of the HML3D dataset or the problem of text-to-motion base model MDM, not our problem. Our framework is agnostic to the data format, and our main contribution is zero-shot interaction generation in the first time. Currently, our method is instantiated on MDM model architecture and follow HML3D data format. Yet, such framework and our main contribution could also be implemented on other base model with different data formats.  We want to thank the reviewer for such constructive suggestion, but we think it is better to try it as an individual paper in our future work.

---

> > ### Comment · Reviewer_VAgJ · 2024-08-11
> > **Response**
> >
> > - Thanks for the clarification about the teaser.
> > - Thanks for the explanation about InterGen.
> > - The HML3D dataset provides the timestamps from the AMASS dataset, so it would be easy to extract them and use SMPL rotations in your setup e.g., following this [repo](https://github.com/Mathux/AMASS-Annotation-Unifier), which has been adopted in other works such as Motion-X. Using body surfaces to study interactions is a clearly more reasonable and correct choice. Also, since you were trying to generate zero-shot interactions, it would be very meaningful to retrain using rotations from AMASS and Motion-X to retrain MDM and use this as a frozen copy.
> > - The limitations of LLM plans should be discussed and included in the final paper.
> > - IK-guidance is a common trick to improve results and it would be nice, if not added to OmniControl, to see your output without it. Also, you referred to slower inference. How does this compare with OmniControl, eg. their guidance vs your IK guidance?

---

> ### Author Response · Authors · 2024-08-11
>
> Thanks for your feedback!
> ### Revise paper
> We will add more clarification of comparison with InterGen and failure cases of LLM plans in our final version.
> ### About IK Guidance
> The quantitative results without IK guidance could also be found in Table 4 of our Appendix. Qualitatively, we find that Motion ControlNet alone (without IK guidance) could lead to good motion generation quality (e.g., good FID). Yet, the joint alignment to expected location is not good enough, which will be vital for our interaction generation process. For example, two people's hands cannot reach exactly the same location when they shake hands. Thus, IK guidance is one of our major contribution in controllable motion generation. Besides, the inference speed comparison with omnicontrol has also discussed in the line 731-739 in our appendix.
> ### Mesh representations
> I agree that 'using body surfaces to study interactions is a clearly more reasonable', especially for interaction modeling. Yet, rethinking the data format of a widely adopted motion generation dataset seems to be beyond the scope of our paper. Our main contribution of this paper is also not the base model (i.e., how to learn a distribution on a specific data format). As I mentioned in the earlier response, many previous text-to-motion base models such as MDM, PriorMDM, T2M-GPT, MLD adopts the same data representation in HML3D, indicating that the data format proposed in HML3D is widely accepted by this community. As our proposed network is a controllable motion generation network which requires such base model as an initialization, training a new base model seems to be beyond this paper's scope. Besides, InterGen also adopts joint locations, instead of original SMPL rotation format or meshes. It will be better to submit an individual paper to rethink the data format of HML3D and InterGen's dataset.
> We will add the discussion of TUCH and STMC paper in our final version. We will also add discussion of motion data representation in the future work section to provide valuable information for this community. We think it will be better to leave the rethinking of motion data format to the future work as an individual paper, as our current paper has already involved many information about the zero-shot interaction generation framework (24 pages without the NeurIPS checklist).
>
> Please raise further questions if you still have concerns about our explanation. Please also consider to raise the rating if our explanations resolve your concerns. Thanks again for your effort to review our paper!

---

### Author Rebuttal · Authors · 2024-08-07

We sincerely thank reviewers’ effort for our paper and the insightful review for us to improve our paper. We carefully read all reviews and address common concerns point by point here.

**Q1@VAgJ, HwRy and 7GL6**: Comparison with InterGen.

**A1**: Firstly, our method is fundamentally different from InterGen. As a zero-shot multi-person motion generation method, InterControl does not learn data distributions from multi-person interaction datasets. This approach offers a significant advantage: the ability to generate interactions for groups of three or more people, as demonstrated in our qualitative results. In contrast, InterGen is inherently limited to generating interactions between only two people. Furthermore, InterControl exhibits greater sensitivity to specific distances, accurately representing scenarios such as physical contact (distance = 0) or maintaining a particular separation (e.g., distance ≥2m). InterGen, however, often struggles to ensure desired distances, particularly in close-contact scenarios like hand-holding. A qualitative comparison between InterControl and InterGen can be found in Figure 1 of the accompanying PDF.

**Q2@VAgJ and HwRy**: Comparison with OmniControl.

**A2**: Firstly, OmniControl does not propose the framework of zero-shot interaction generation, thus our focus is fundamentally different from it. We were unaware of OmniControl when we chose controllable single-person motion generation to achieve the goal of zero-shot interaction generation. The detailed comparison with OmniControl can be found in Sec. A.4 in the Appendix. Our major advantage over OmniControl in single-person motion control is the superior performance on non-root joint control. Secondly, we have included a quantitative comparison with OmniControl on single-person motion evaluation in Table 1 of our paper. To address the concern of HwRy, we also include a quantitative comparison with OmniControl on zero-shot interaction generation using our collected contact plans in the following Table. OmniControl exhibits poorer joint control performance in the contact joint pairs. Thirdly, qualitative comparison with OmniControl can be found in Figure 2 of the PDF. It also demonstrates cases where the joint alignment in contact joints (e.g., distance = 0) is not as precise as ours.

| Method | Traj. err. (20 cm) ↓ | Loc. err. (20 cm) ↓ | Avg. err. (m) ↓ |
| --- | --- | --- | --- |
| PriorMDM [51] | 0.6931 | 0.3487 | 0.6723 |
| OmniControl [65] | 0.0322 | 0.0029 | 0.0194 |
| Ours | 0.0082 | 0.0005 | 0.0084 |


**Q3@VAgJ, HwRy and 7GL6**: Interpenetration (body penetration, human collision) issues.

**A3**: As our method is a kinematics-based motion generation approach, it primarily focuses on semantic alignment with language rather than local penetration or artifacts, which is consistent with MDM and other kinematics-based methods. The main evaluation metrics for single-person motion generation are FID or Top-3 classification precision, which measure the ability to learn data distribution. Consequently, local artifacts are not the primary concern of kinematics-based motion generation methods, and eliminating such artifacts is challenging for these approaches. For instance, InterGen also exhibits body penetrations (e.g., "Two people embrace each other" in https://tr3e.github.io/intergen-page/). In a fair comparison with other zero-shot methods, e.g., PriorMDM, we demonstrate superior results in body penetration avoidance. Our IK guidance ensures that any two characters' torso joints maintain a minimum distance of one meter. PriorMDM displays significant torso collision in the qualitative results (refer to Figure 3 in our paper ), while our method only shows minor collisions of limbs.

Moreover, if we need to eliminate penetration issues in specific applications, physical animation in a simulator presented in our application section (Lines 334-342 and Figure 1(c)) will be an effective solution. This is effective because the simulator inherently prevents collisions. Our qualitative results on physical animation demonstrate that training-free imitation learning methods [40] execute our generated interactions effectively, a point acknowledged by Reviewer#HwRy.

**Q4@HwRy and 7GL6**: Unsmooth motion (abrupt velocity changes) in qualitative results.

**A4**: Thank you for the detailed review. We set the FPS to 15 in the visualization video for qualitative results, while the common practice is 30. It may misleadingly suggest that our generated motions are not smooth enough. Additionally, some cases in the visualization videos use more than 5 contact pairs and short intervals between contact and separation, potentially leading to higher velocities in these intervals. As explained in our paper, IK guidance can adaptively modify velocities from the start to frame n. Generated motion will be smoother if the frame number n is larger. However, some cases in the visualization use a very small n, which could be easily misinterpreted. We sincerely appreciate the insightful review and will carefully revise it in our final version.

Furthermore, we have observed that many motion generation methods (e.g., InterGen) use a 1D Gaussian filter to smooth the generated motion without affecting the semantics. In the following Table, we present quantitative results of acceleration in generated motions where both methods adopt the same post-processing. Our acceleration is similar to InterGen's when utilizing the standard 1D Gaussian filter as post-processing, demonstrating that our velocity changes are comparable to existing methods'. Additionally, we provide a sequence of acceleration in two people fighting to qualitatively compare the acceleration between our generated interaction and InterGen's. This figure can be found in Figure 3 of the attached PDF. It further illustrates that our acceleration is comparable to InterGen's.

| Method | Acceleration (m/frame^2) |
| --- | --- |
| InterGen [36] | 0.0046 |
| Ours | 0.0042 |

---

> ### Author Response · Authors · 2024-08-12
>
> Dear Reviewers,
>
> We are happy to address any additional concerns you may have. If, after our discussion, you find that all your concerns have been resolved, we would greatly appreciate your consideration in updating the final rating.
>
> Thank you for your time and expertise.
>
> Sincerely,
>
> The Authors

---

### Decision · Program_Chairs · 2024-09-25

**Decision:**

Accept (poster)

**Comment:**

Post-rebuttal, reviews were split across borderline, with one in favor of acceptance, two borderline accept, and one borderline reject. In post-rebuttal discussions, the borderline reject reviewer commented that they would be in favor of acceptance if the other reviewers were too (which they are), leading to all accept recommendations. The AC examined the paper, the reviews, the rebuttal and the discussion and is in favor of accepting the paper: this work will be of interest to the community. While there are some lingering concerns that might be fixed in to-be-seen revision, none are show-stoppers

Since the rebuttal and ensuing discussion was particularly important for the acceptance of the paper, AC strongly urges the authors to incorporate the rebuttal and feedback from the discussion into the final version of the paper. In particular, the teaser should be changed to remove images (which may give a misleading impression), and the final version of the paper should use the extra page to also have a discussion about mesh-based representations and body surfaces. While there is no mechanism to review the paper, the AC would remind the authors that other readers may have similar reactions, and so incorporating the results will also maximize the impact of the paper.